# Meta-analysis of neural systems underlying placebo analgesia from individual participant fMRI data

Matthias Zunhammer [1], Tamás Spisák[1], Tor D. Wager [2✉], Ulrike Bingel[1✉] & The Placebo Imaging Consortium*

The brain systems underlying placebo analgesia are insufficiently understood. Here we performed a systematic, participant-level meta-analysis of experimental functional neuroimaging studies of evoked pain under stimulus-intensity-matched placebo and control conditions, encompassing 603 healthy participants from 20 (out of 28 eligible) studies. We find that placebo vs. control treatments induce small, widespread reductions in pain-related activity, particularly in regions belonging to ventral attention (including mid-insula) and somatomotor networks (including posterior insula). Behavioral placebo analgesia correlates with reduced pain-related activity in these networks and the thalamus, habenula, mid-cingulate, and supplementary motor area. Placebo-associated activity increases occur mainly in frontoparietal regions, with high between-study heterogeneity. We conclude that placebo treatments affect pain-related activity in multiple brain areas, which may reflect changes in nociception and/or other affective and decision-making processes surrounding pain. Between-study heterogeneity suggests that placebo analgesia is a multi-faceted phenomenon involving multiple cerebral mechanisms that differ across studies.

---

[1] Center for Translational Neuro- and Behavioral Sciences, Dept. of Neurology, University Hospital Essen, Essen, Germany. [2] Cognitive and Affective Neuroscience Laboratory, Department of Psychological and Brain Sciences, Dartmouth College, Hanover, NH, USA. *A list of authors and their affiliations appears at the end of the paper. ✉email: tor.d.wager@dartmouth.edu; ulrike.bingel@uk-essen.de

Placebo effects contribute substantially to treatment outcomes in both medical research and clinical practice. A better understanding of the underlying mechanisms is thus important for optimizing drug development and clinical care[1]. Placebo analgesia is the most robust and best-studied type of placebo effect[1–3]. A growing number of neuroimaging studies elucidate the brain correlates of placebo analgesia. These studies, and meta-analyses of their findings, have provided evidence for the involvement of brain regions linked to nociceptive processing, including early pain-gating mechanisms, but also to decision-making, cognitive appraisal, reward/motivation, emotional regulation[4–7], and other forms of learning and social cognition[8] relevant for health behaviors.

Nevertheless, the results of these studies vary substantially[4], and the lack of large-sample assessments hampers the detection of small to moderate effects[9] and makes it difficult to identify precisely which structures are consistently altered by placebo treatment. Previous meta-analyses have all relied on published coordinates of activation peaks. These incomplete summaries of the full activation maps provide only approximate information on replicability across studies and are susceptible to bias[10]. These limitations can be overcome by meta-analyses based on single-participant, whole-brain images, which are sometimes referred to as 'mega-analyses'[11]. As meta-analyses on participant-level data are preferable in terms of statistical power and risk-of-bias[11], a mega-analysis of placebo-induced brain activity can be expected to foster convergence in our understanding of placebo analgesia, to provide novel insights into the underlying neural mechanisms, and guide the development of predictive methods of individual placebo analgesia from neuroimaging data, which would be of crucial importance both from a clinical and drug development point of view. Here, we conducted a systematic participant-level meta-analysis of 20 independent neuroimaging studies on experimental placebo analgesia. Based on whole-brain activation patterns in a total of $N = 603$ healthy participants, we mapped the effects of placebo treatment on pain-related brain activity and identified neural correlates of individual differences in behavioral placebo analgesia.

## Results

**Image quality**. All included studies ($N = 603$ participants from 20 studies, see ref. [7] and Table 1) aimed at covering the whole brain down to the mid-pons/superior cerebellar level. Image alignment to MNI-space was satisfactory for all studies, but coverage was often incomplete near the boundaries of the brain (see Supplementary Figs. 2 and 3), particularly in the inferior brainstem, cerebellum, and ventral prefrontal regions. These partially missing data are likely due to between-study differences in field-of-view and/or signal dropout artifacts. For one study[12], only maps with white-matter regions masked out were available.

Outlier screening (see Supplementary Methods and Results) indicated pain ratings that were too low for inclusion in six participants (responses <5% of the pain scale). Problematic image features were found in 12 (2.0%) participants. These include imaging artifacts (six participants), extreme values (four participants), or likely errors in first-level analysis (two participants). These outlier participants were retained in our analysis (see Supplementary Methods and Results, Supplementary Figs. 4, 5, Supplementary Table 8 for an analysis excluding high risk-of-bias studies and outlier participants, which shows similar results to the full sample analysis in terms of effect sizes).

**Voxel-wise results: pain stimulation effects**. Painful stimulation compared to baseline induced large peak effects ($g > 0.8$); with the largest located in the insula, bilaterally (Fig. 1a, Supplementary Fig. 6, Supplementary Table 9). In general, cerebral activations

and de-activations were found in regions typical for experimental pain (compare: Fig. 1b and ref. [13]). The $\tau$-statistic indicated considerable between-study heterogeneity in pain-related activity throughout most of the brain (Supplementary Fig. 7), which was expected given the large inter-study diversity regarding experimental pain-induction, image acquisition, and image processing (Supplementary Tables 2–5).

**Voxel-wise results: effect of placebo treatment**. In general, placebo treatment had a small ($g < 0.2$) effect on pain-related brain activity, as compared to the matched control conditions (Fig. 2a, b). Significant placebo-associated decreases were found in the right insula, near the habenula and the splenium of the corpus callosum, and in the cerebellum ($p < 0.05$, FWER corrected with pTFCE; Fig. 2a light blue, Supplementary Fig. 8, Supplementary Table 10). No areas showed placebo-related increases at the FWER-corrected threshold treating study as a random effect.

Estimated between-study heterogeneity in voxel-level effect sizes was low in the significantly de-activated regions (Fig. 2c, Supplementary Fig. 8, Supplementary Table 10). However, many regions of the brain showing sub-threshold placebo-related increases showed statistically significant $\tau$-values, indicating between-study heterogeneity in effects (Fig. 2c). These included multiple prefrontal cortical areas, perigenual anterior cingulate cortex, intraparietal sulcus, precuneus, basal ganglia, and the left middle insula. A brain-wide correlation analysis indicated that placebo treatment effects were positively and significantly correlated across brain regions with $\tau$-values ($r = 0.191$, 95% CI [0.187, 0.196], $p < 0.001$, Supplementary Fig. 9), indicating that areas showing placebo-induced increases tended to have higher levels of between-study heterogeneity. Thus, activation increases varied more substantially across studies than activation decreases.

We therefore performed an exploratory fixed-study-effects analysis of placebo effects, which tests for effects within this set of studies without the intent of generalizing to new, unobserved studies. In addition to decreases reported above, this analysis showed reduced activity in the middle cingulate cortex, the bilateral supplementary motor area (SMA), left fusiform cortex and cerebellum (Fig. 2d, light blue). The fixed-effects analysis revealed significant placebo-induced activation in the anterior intraparietal sulcus, precuneus, and dorsolateral prefrontal cortex (DLPFC) (Fig. 2d gold, Supplementary Table 11).

To further follow up on potential sources of between-study heterogeneity, we explored the possibility of explaining heterogeneity through study-level experimental features, such as the method of placebo induction. A preliminary comparison of placebo induction methods (conditioning and suggestions versus suggestions only) showed no significant differences in placebo-related brain activity after correction for multiple comparisons (Supplementary Fig. 10).

**Voxel-wise results: correlations with placebo analgesia**. In the vast majority of voxels (Fig. 3a), placebo analgesia was negatively correlated with placebo-induced changes in brain activity. Thus, the larger the activity decreases, the more analgesia a participant reported (Fig. 3a, blue and light blue). Negative correlations were strongest and statistically significant in the bilateral thalamus, right anterior, middle and posterior insula, right secondary somatosensory cortex, right superior temporal gyrus, right cerebellum (around the dorsal part of lobule VI), basal ganglia, the mid-cingulate cortex, as well as SMA/pre-SMA ($p < 0.05$, FWER corrected with pTFCE; Fig. 3a light blue; also see Supplementary Fig. 11, Supplementary Table 12). The activity of contralateral (left) areas of the insula ($z$-score $= 3.9$, $r = 0.17$, $p = 0.00005$), the

**Table 1 Studies included in the meta-analysis.**

| | First author | Year | *n* | Design | Mean age (y) | Sex (% male) | Pain stimulus | Placebo induction | Treatment |
|---|---|---|---|---|---|---|---|---|---|
| 1 | Atlas[48] | 2012 | 21 | within | 25 | 48 | heat | sug | IV-infusion |
| 2 | Bingel[53] | 2006 | 19 | within | 24 | 79 | laser | sug + cond | topical cream |
| 3 | Bingel[70] | 2011 | 22 | within | 28 | 68 | heat | sug + cond | IV-infusion |
| 4 | Choi[54] | 2011 | 15 | within | 25 | 100 | electrical | sug + cond | IV-infusion |
| 5 | Eippert[56] | 2009 | 40 | within | 26 | 100 | heat | sug + cond | topical cream |
| 6 | Ellingsen[71] | 2013 | 28 | within | 26 | 68 | heat | sug | nasal spray |
| 7 | Elsenbruch[72] | 2012 | 36 | within | 26 | 42 | distension | sug | IV-infusion |
| 8 | Freeman[51] | 2015 | 24 | within | 27 | 50 | heat | sug + cond | topical cream |
| 9 | Geuter[55] | 2013 | 40 | within | 26 | 100 | heat | sug + cond | topical cream |
| 10 | Kessner[73] | 2013 | 39 | between | 26 | 51 | heat | cond | topical cream |
| 11 | Kong[49] | 2006 | 10[c] | within | 27 | 60 | heat | sug + cond | sham acupuncture |
| 12 | Kong[50] | 2009 | 12[a] | within | 26 | 42 | heat | sug + cond | sham acupuncture |
| 13 | Lui[74] | 2010 | 31 | within | 23 | 45 | laser | sug + cond | sham TENS |
| 14 | Rütgen[52] | 2015 | 102 | between | 25 | 31 | electrical | sug + cond | pill |
| 15 | Schenk[75] | 2014 | 32 | within | 26 | 53 | cap + heat | sug | topical cream |
| 16 | Theysohn[76] | 2014 | 30 | within | 35 | 50 | distension | sug | IV-infusion |
| 17 | Wager[12,A] | 2004 | 24 | within | NA | NA | electrical | sug | topical cream |
| 18 | Wager[12,B] | 2004 | 23 | within | NA | NA | heat | sug + cond | topical cream |
| 19 | Wrobel[57] | 2014 | 38 | within | 26 | 58 | heat | sug + cond | topical cream |
| 20 | Zeidan[43] | 2015 | 17[a] | within | 28 | 47 | heat | sug + cond | topical cream |

*A* Sub-study 1, *B* Sub-study 2, *between* between-group design, *cap*, capsaicin, *cond* conditioning, *IV* intravenous, *L* left, *NA* not available, *R* right, *sug* suggestions, *TENS* transcutaneous electrical nerve stimulation, *within* within-subject design.
[a]Placebo-treatment groups, only.

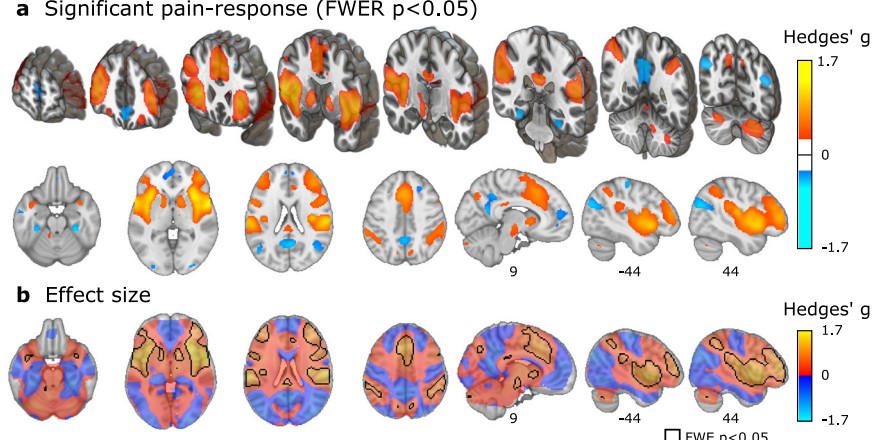

**Fig. 1 Pain-related activity in experimental placebo imaging studies. a** Statistically significant pain-responses (permutation test, controlled for FWER, two-sided *p* < 0.05), and **b** whole-brain unthresholded standardized effect size *g*, of acute pain stimulation > baseline, pooled across placebo and control conditions (FWER-corrected permutation test results are delineated as a back contour); range *g*: [−0.82, 1.68]; all: *n* = 543–603 individuals from 17 to 20 independent studies per voxel. Three-dimensional coronal slices are equidistantly distributed from *y* = 60 to −68 mm. Axial slices range equidistantly from *z* = −22 to 42 mm. Custom coordinates for sagittal slices are displayed in mm. Source data (results as 3d-volumes) are provided at https://osf.io/n9mb3/.

secondary somatosensory cortex (*z*-score = 4.0, *r* = 0.21, *p* = 0.00003) was also negatively correlated with analgesia but was only significant without correcting for multiple comparisons (Fig. 3a, blue).

Positive correlations between behavioral placebo analgesia and brain activity, i.e. increasing brain activity with stronger placebo response, did not reach statistical significance (*p* < 0.05, FWER corrected, pTFCE). Without correcting for multiple comparisons, positive correlations (Fig. 3a, b, red) were observed near the subgenual cingulate cortex (*z*-score = 1.8, *r* = 0.14, *p*_uncorr = 0.036), in the orbitofrontal cortex (*z*-score = 2.7, *r* = −0.17, *p*_uncorr = 0.003), and the prefrontal pole (*z*-score = 2.6, *r* = −0.13, *p*_uncorr = 0.005).

Levels of between-study heterogeneity were negligible in regions showing significant correlations between behavioral placebo

analgesia and brain activity (Fig. 3b, Supplementary Fig. 11, Supplementary Table 12), suggesting that correlations were driven by inter-individual differences rather than systematic differences across studies. Across the brain, between-study heterogeneity did not reach FWER significance, but was largest in the basal ganglia, orbitofrontal and dorsolateral prefrontal cortices; see Fig. 3c). Between-study heterogeneity was not spatially associated with correlations across voxels (Supplementary Fig. 12); thus, the most heterogeneous regions were not those with the strongest effects.

**Network- and region-based results: effects of painful stimulation.** Activation for painful stimulation compared to baseline (averaged across placebo and control conditions) showed activation of multiple expected cortical and subcortical regions

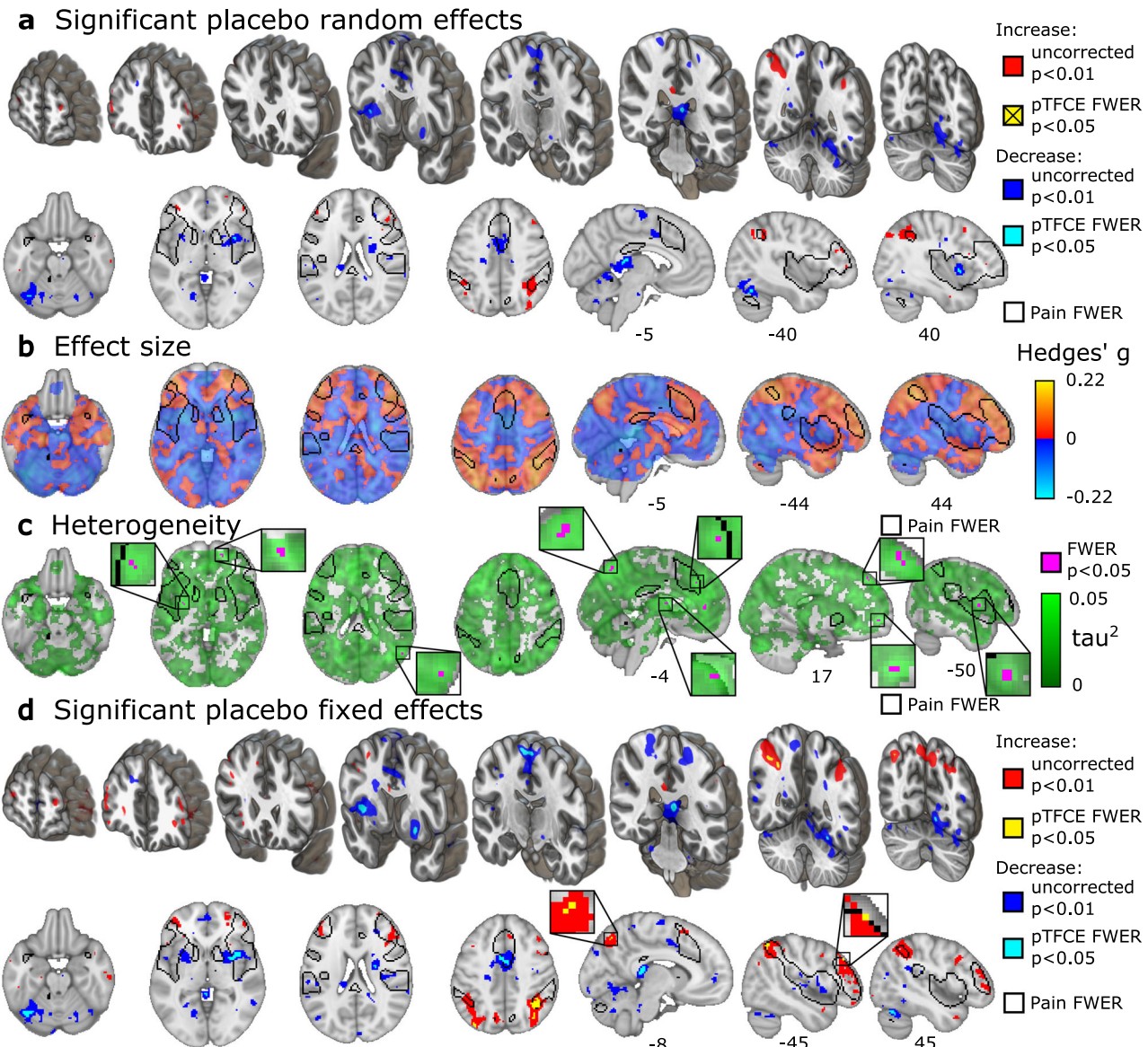

**Fig. 2 Placebo-induced changes in pain-related activity. a** Areas of statistically significant placebo treatment effect, assuming random study-effect, thresholded according to z-tests (uncorrected for multiple comparisons, two-sided $p < 0.01$, red and blue) and thresholded according to pTFCE-enhanced permutation test (controlled for FWER, two-sided $p < 0.05$, light blue, activity increases did not reach statistical significance); **b** unthresholded standardized effect size $g$ of placebo treatment effect (range: [−0.19, 0.17]); **c** between-study heterogeneity $\tau$ (range: [0, 0.43]) with permutation test results (controlled for FWER, one-sided, $p < 0.05$, green; $\tau$ is plotted as $\tau^2$ to emphasize regions of high heterogeneity. **d** significant placebo-effects assuming fixed study-effect (range: [−0.22, 0.22]) thresholded according to z-tests (uncorrected for multiple comparisons, two-sided $p < 0.01$, red and blue) and thresholded according to pTFCE-enhanced permutation test (controlled for FWER, two-sided $p < 0.05$, light blue and gold); all: $n = 543$ to 603 individuals from 17 to 20 independent studies per voxel. **b**, **c**, and **d** are shown with a contour of FWER-corrected permutation test results for pain > baseline, as shown in Fig. 1a. Small FWER-corrected clusters are zoomed in insets. Three-dimensional coronal slices are equidistantly distributed from $y = 60$ to −68 mm. Axial slices range equidistantly from $z = -22$ to 42 mm. Custom coordinates for sagittal slices are displayed in mm and were chosen to highlight important areas of activation. Source data (results as 3d-volumes) are provided at https://osf.io/n9mb3/.

(Fig. 4, Column 2). These included activation in the ventral attention network (which encompasses the insulae), the fronto-parietal network[14], and the somatomotor network. Positive associations were also found in all insular sub-regions and most thalamic nuclei, including the intralaminar nuclei targeted by ascending nociceptive pathways, the mediodorsal 'limbic association' nucleus, and the ventro-basal complex, including the ventro-posterior lateral (VPL) nucleus[15]. A tendency towards negative associations was found for the lateral geniculate, medial geniculate, and pulvinar nuclei, which are known for their predominantly visual and auditory roles[15].

**Network- and region-based results: effect of placebo treatment**. Network similarity analysis indicated that placebo treatment reduced activity in the ventral attention and the somatomotor networks[14]; (Fig. 4, Column 3) which includes the mid-cingulate cortex (localized particularly to area 24pr in ref. [16]). In the insula, placebo reduced activity in bilateral middle short gyrus and right posterior short gyrus, corresponding to the dorsal anterior/mid-insula, as well as a trend towards reduced activity in the right anterior long gyrus (posterior insula, contralateral to stimulation in most studies). Thalamic nuclei showed tendencies towards placebo-induced decreases in areas strongly activated in pain. The

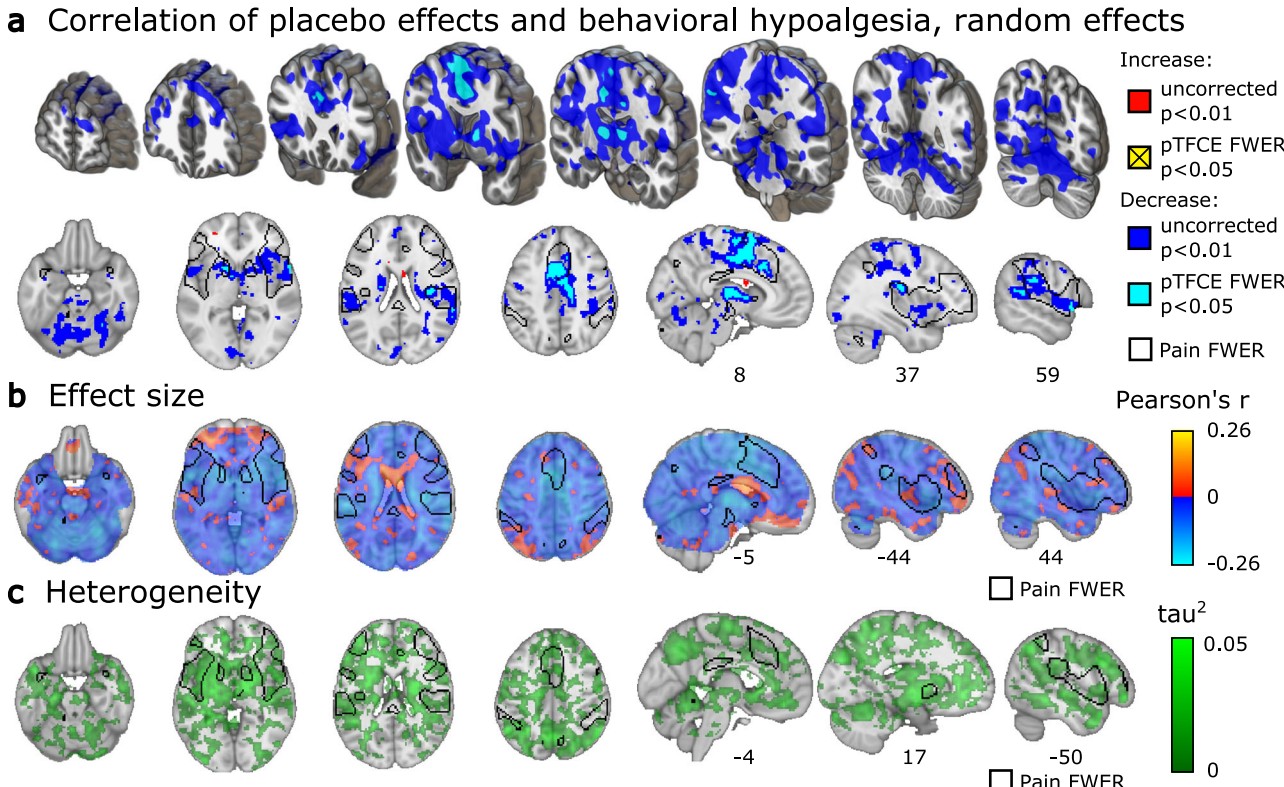

**Fig. 3 Correlations of behavioral placebo analgesia and changes in pain-related brain activity. a** Whole-brain areas of statistically significant correlation (Pearson's $r$) between behavioral placebo analgesia ($pain_{control} − pain_{placebo}$) and placebo-related activity changes ($pain_{placebo} − pain_{control}$), thresholded according to $z$-tests (uncorrected for multiple comparisons, two-sided $p < 0.01$, red and blue), and thresholded according to pTFCE-enhanced permutation test (controlled for FWER, two-sided $p < 0.05$, light blue, increased correlations did not reach corrected statistical significance); **b** unthresholded Pearson's $r$, range: [−0.26; 0.17]; **c** between-study heterogeneity $\tau$ (range: [0, 0.32]) for the same relationship (permutation test controlled for FWER, one-sided $p < 0.05$, indicated no statistically significant voxels); $\tau$ is plotted as $\tau^2$ to emphasize regions of high heterogeneity. all: $n = 384–460$ individuals from 15 to 18 independent studies per voxel. **a**, **b**, and **c** are shown with a contour of FWER-corrected permutation test results for pain > baseline, as shown in Fig. 1a. Correlations were computed across individual participants in the full sample, excluding between-group studies (where individual estimates of behavioral placebo analgesia are not possible). On panels **a** and **b**, red-yellow and blue-light blue shades denote increased and decreased activity associated with larger placebo analgesia, respectively. Three-dimensional coronal slices are equidistantly distributed from $y = 60$ to $−68$ mm. Axial slices range equidistantly from $z = −22$ to $42$ mm. Custom coordinates for sagittal slices are displayed in mm. Source data (results as 3d-volumes) are provided at https://osf.io/n9mb3/.

strongest decreases were found in the VPL, a primary target of the spinothalamic tract, and the habenula. Observed placebo effects in other regions tended to be smaller.

**Network- and region-based results: correlations with placebo analgesia**. As with the main effects of placebo vs. control, network similarity-based analysis of regions correlated with placebo analgesia indicated that activity in the ventral attention and somatomotor networks was negatively correlated with behavioral placebo responses (Fig. 4), i.e., strong placebo responders showed larger deactivation with placebo treatment. Within the right insula, several regions tended towards negative correlations with placebo responses, especially the anterior long gyrus (posterior insula), middle short gyrus (dorsal anterior/mid insula), and anterior inferior cortex (ventral insula). In the thalamus, stronger placebo analgesia was correlated with reductions in multiple thalamic regions, including all seven regions that responded strongly to pain in this sample (intralaminar, ventrolateral, ventral anterior, ventromedial, mediodorsal, antero-medial, and anterio-ventral nuclear groups), and thalamic targets of the spinothalamic tract (ventro-posterior-lateral [VPL] and -medial [VPM]).

## Discussion

In this collaborative effort, we performed a comprehensive, large-scale ($N = 603$) participant-level voxel-based neuroimaging meta-analysis of placebo analgesia, involving the majority of eligible experimental neuroimaging studies. Our results provide a reliable, aggregated view of the size, localization, significance, and heterogeneity of placebo-effects on pain-induced brain activity. In a previous paper, we focused on the question of whether placebo analgesia involves changes in the neurologic pain signature (NPS)[17], a machine-learning based weighted, multi-voxel summary metric (covering about 10% of the brain), that can be interpreted as a neuromarker of nociceptive pain. This previous study revealed that behavioral placebo analgesia was associated with significant but small effects in the NPS, pointing to the relevance of other brain areas and networks. Accordingly, characterizing this potentially broader set of changes was the key focus of this voxel-wise whole-brain investigation (for a comparison with regions involved in the NPS, see Supplementary Fig. 13). The present results corroborated previous findings of increases in frontal-parietal regions and reductions in the insula. In addition, they revealed new effects systematically missed in previous smaller-scale analyses, including reductions in the habenula, specific parts of the thalamus (particularly VPL, a

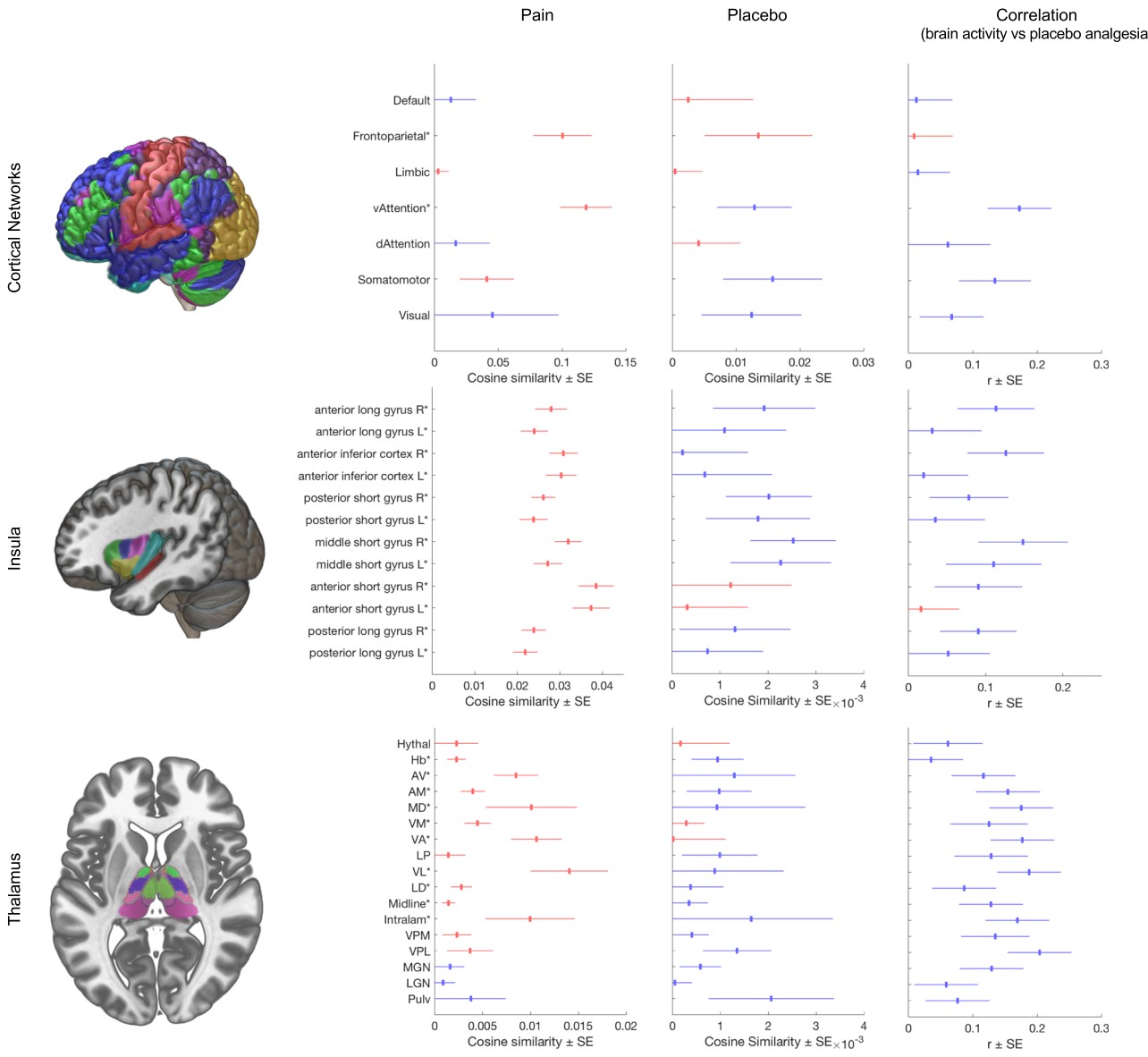

**Fig. 4 Similarity-based analysis of brain activity in functional cortical networks, insula, and thalamus.** Column 1 (header): Depiction of atlases: Row 1: whole-brain cortical networks of functional connectivity[14], Row 2: insular sub-regions[66] Row 3: thalamic nuclei[67]. See Supplementary Fig. 14 for further details. Column 2: Mean (±SEM) cosine similarity (c) of pain-related activity, n = 603 from 20 independent studies; Column 3: Mean (±SEM) cosine similarity (**c**) of placebo-induced changes in pain-related activity (Column 2); all: n = 603 from 20 independent studies. Column 4: Correlation (Pearson's r ± SEM) between behavioral placebo response and cosine similarity estimates of placebo-related activity; n = 460 from 18 independent studies. In Columns 2 and 3, red and blue colors denote increased and decreased pain-related activity, respectively. In Column 4 red and blue shades denote increased and decreased activity associated with larger placebo analgesia, respectively. Asterisks (*) denote significant differences from zero, according to two-sided t-tests (p < 0.05, uncorrected for multiple comparisons). Source data (results as 3d-volumes) are provided at https://osf.io/n9mb3/. Hythal Hypothalamus, Hb Habenular, AV anterior ventral, AM anterior medial, MD mediodorsal, VM ventral medial, VA ventral anterior, LP lateral posterior, VL ventral lateral, LD lateral dorsal, Intralam intralaminary, VPM ventral posterior medial, VPL ventral posterior lateral, MGN Medial Geniculate Nucleus, LGN Lateral Geniculate Nucleus, Pulv Pulvinar.

nociceptive nucleus), and the cerebellum, promising new targets in explaining placebo analgesia.

Here, we discuss our results in correspondence to two key open questions: (i) how strongly do placebo treatments affect the same systems involved in nociception and pain generation (as indicated by the contrast control > placebo); and (ii) which systems are engaged by placebo treatments (as indicated by the contrast placebo > control) and may therefore reflect top-down modulatory mechanisms.

Our study-as-random-effect results provide strong evidence for placebo-associated reductions of pain-related activity in some brain areas linked to nociception and pain and indicate that these are generalizable across experimental paradigms. We also provide strong evidence that the degree to which pain-related activity is reduced in these brain areas correlates with the magnitude of behavioral hypoalgesia across individuals.

Specifically, the placebo-associated decrease and its correlation with behavioral pain ratings were most prominent in regions

located within the ventral attention and the somatomotor cortical networks, including the anterior insula and targets of the spinothalamic tract[18], including the thalamic VPL complex the posterior insula and, moreover, in the habenula (Figs. 2b and 3b; Fig. 4, Column 3 and 4). Correlations were additionally pronounced in both lateral (VL and VPM/VPL) and medial (intralaminar and MD) thalamic nuclei (Fig. 4). These regions are targets of ascending nociceptive systems and, as expected, were also strongly engaged during painful stimulation. In summary, placebo-associated down-regulation seems to affect thalamocortical pathways related to nociception and pain[13,19], particularly in strong placebo responders. The relatively low between-study heterogeneity in these regions indicates that variability in placebo-related reductions is primarily a function of the individual responder rather than the paradigm used.

These findings complement previous findings of small but statistically significant placebo-induced reductions in the NPS[7]. In this previous study, NPS reductions also correlated with the magnitude of placebo analgesia. Here, findings of VPL reductions with placebo and widespread correlations between analgesia and correlations in broadly pain-related systems support the conclusion that alterations in nociception and pain construction are an important element of placebo analgesia. The small effect sizes however indicate that nociceptive changes are unlikely to be a complete explanation. The strengths of the previous NPS findings were that the neuromarker was identified independently from the present sample and validated in over 40 published cohorts to date. However, limitations were that (1) the NPS may not perfectly characterize nociceptive processing in this sample, and some nociceptive pain-related effects may be missed; (2) it cannot provide a broad view of effects across the brain, including areas like the habenula and many cortical areas; and (3) it tests a distributed pattern response and is not suited to identify placebo effects in VPL (or any other region) in particular. The present findings of effects in specific thalamic and other regions are complementary and important, in that they provide a region-level inference about effects in neuroanatomically defined nociceptive pathways. Thus, overall, we believe that placebo effects affect the circuitry involved in pain generation to some degree, in a manner that varies across individuals more than across studies in the present dataset, but also includes other brain effects beyond nociception that may be important for the emotions, decision-making, and behaviors surrounding pain.

Placebo-related decreases were not restricted to pathways associated with nociception. Brain regions traditionally linked to self-regulation and high-level action selection, particularly the SMA[20–25] also showed reduced activity during placebo analgesia, particularly in strong placebo responders (Figs. 2 and 3). Thus, it is possible that some of these effects reflect shifts in motivation and decision-making in the context of pain. These findings extend previous meta-analyses, which all highlighted deactivations in the mid-cingulate, but not the SMA or pre-motor areas[4–6]. In addition to action planning and self-regulation, they may be related to other cognitive operations related to evaluating pain under placebo treatment, including error monitoring, prediction errors, and sequence processing[20,26].

One of the strongest effects was found in the left putamen, which de-activated with placebo in proportion to analgesia. This is in line with multiple studies reporting correlations between placebo analgesia and both fronto-parietal and limbic fronto-striatal pathways[8] and might be related to the (prefrontal) suppression of striatal prediction errors or other aversive circuits[27].

Interestingly, each of the main analyses revealed prominent placebo-related reductions in cerebellar regions. While this in line with some previous findings (e.g.[12]), cerebellar effects were not reported in previous meta-analyses of placebo analgesia, possibly

due to insufficient cerebellar coverage across studies. Here, the dorsomedial cerebellum showed a profile of responses to painful events, reductions with placebo, and correlations with the magnitude of placebo analgesia. Some cerebellar regions have been linked to pain, and others to other cognitive, affective, and motor processes[28,29] and patients with cerebellar infarctions show reduced placebo analgesia[30]. Cerebellar reductions and correlations are centered in vermis areas V and (to a lesser degree) II[31–33], which are associated with somatomotor and limbic cortical networks, respectively. Thus, the best interpretation of cerebellar effects here is that they are related partly, but not exclusively, to somatomotor networks and pain. Placebo hypoalgesia-related activity changes in the VL nucleus, a target of cerebellothalamic tract[15], also suggest that fronto-cerebellar connectivity may pose a promising novel target for future in-depth studies on the mechanisms of placebo analgesia.

In contrast to these placebo-related deactivations, some regions displayed increased pain-related activity as an effect of placebo treatment, which is often interpreted as participating in the construction of top-down representations of context (including beliefs and expectations). These regions tend to be localized in the frontoparietal network (FPN). These increases were statistically significant only in the fixed effect analysis and involved the right DLPFC (with subthreshold activation on the left side), as well as the bilateral intraparietal sulcus. While the fixed-effect analysis provides limited generalizability, this result is very much in line with previous neuroimaging studies highlighting the importance of the DLPFC in initiating and maintaining the top-down effect of treatment expectation on nociceptive processing and pain. E.g. activity in the DLPFC precedes and scales with activity changes in downstream pain modulatory areas and prevents the extinction of once learned placebo analgesia[27]. Moreover, transient inhibition of the DLPFC using transcranial magnetic stimulation led to reduced placebo analgesia[34].

The lack of FWER-corrected increases might be attributed to the reduced power in the study-as-random-effect analysis, which is very conservative. However, the heterogeneity analysis indicates that the between-study variance is significantly higher than expected in key fronto-parietal areas (Fig. 2c). This suggests that, in contrast to the consistent placebo-induced decreases across studies, placebo-related increases in brain activity are more heterogenous across placebo induction techniques. For instance, significant heterogeneity in the DLPFC and perigenual ACC might reflect the differing engagement of descending pain regulatory mechanisms across studies, although these regions are clearly not exclusively associated with pain modulation (e.g. see ref. [35]).

Between-study heterogeneity was statistically significant throughout the frontal lobe (Fig. 2c), which may reflect inter-study variation in participants' appraisal of the context and internal responses, e.g., expectations[36]. The degree to which prefrontal systems are required for analgesia may vary. For example, in ref. [37], placebo analgesia was predicted by fronto-parietal activity in regions associated with emotion regulation but not working memory. Emotion regulation, in particular reappraisal strategies involving self-generated positive contexts for experiences, appears to involve fronto-parietal networks in reducing negative affect. Other recent studies have also found correlations between placebo analgesia and DLPFC connectivity[38,39] (e.g., with the nucleus accumbens[40]) and opioid binding in prefrontal cortex[41,42]. By contrast, other studies have found that mindfulness practice can reduce pain without fronto-parietal activation or appreciable deactivation in spinothalamic targets[43]. These strategies focus on acceptance without judgment rather than active re-contextualization, which may be another important component of placebo analgesia.

In sum, placebo analgesia may involve multiple alterations in appraisal systems, reflecting multiple underlying mechanisms[44]. Our results suggest that placebo effects are not restricted solely to either sensory/nociceptive or cognitive/affective processes, but likely involve a combination of mechanisms that may differ depending on the paradigm and other individual factors. Understanding the neural and neurochemical pathways underlying this variability will pave the way to systematically utilize/modulate placebo responses in a context-, patient-, and disease-specific manner. Fostering the therapeutic processes underlying placebo effects in clinical settings promises to boost the efficacy (and tolerability) of analgesic drug treatments. Likewise, controlling and homogenizing placebo responses during drug development can enhance the assay sensitivity in clinical trials. Finally, biomarkers based on the types of brain alterations we identify here, and reported in other studies[38,45], may help to dissect placebo from analgesic drug responses in pre-clinical trials.

The present findings must be interpreted in the light of several limitations. First, as our findings are based on experimental placebo interventions in healthy volunteers, they may not generalize to clinical settings. Second, the present study covered a wide range of experimental placebo paradigms and conditions. This is favorable in terms of establishing the broad generalizability of results, but it also means that findings have to generalize over many sources of variation: Paradigm, population/sample, scanner, and choice of analysis methods. Effect size estimates are thus likely overly conservative compared with what may be possible as analysis methods continue to become standardized and methodological advances reduce inter-subject and inter-study variability. Further, the fact that this meta-analysis was based on participant-level statistical summary images from variety of different pre-processing pipelines (s. Supplementary Table 5) likely had a negative impact on spatial brain mapping accuracy, in particular since different software packages[46] (and therefore MNI-templates[47]) and different spatial smoothing kernels were used. This study therefore trades off spatial precision for generalizability across scanners, populations, pipelines, and paradigms. Finally, some brain regions, notably the orbitofrontal cortex (including the vmPFC), the inferior cerebellum and the top-most part of the brain were not fully covered (Supplementary Figs. 2 and 3) and placebo-related activation changes in these regions could not be assessed and may be missed.

In this systematic meta-analysis of individual participant data, we show that placebo treatments induce small, yet robust, inhibitory effects in large parts of the brain. These involve selected regions within the ventral attention and somatomotor networks, including targets of spinothalamic-afferents strongly linked to nociception, and are consistent across studies. Further, our study corroborates the relevance of placebo-related activity in fronto-parietal areas; however, the degree and relevance of fronto-parietal activity show large between-study heterogeneity. Our results suggest, that placebo is neither restricted to sensory/nociceptive nor to selective cognitive/affective processes but likely involves a combination of mechanisms that may differ depending on the paradigm and other individual factors.

## Methods

The present study is a systematic meta-analysis of participant-level data across 20 published studies. A previous paper on this dataset[7] tested placebo effects on a single, a priori pain-related measure (the neurologic pain signature[17]). Here, we used the same data set to map placebo responses across the brain. In contrast to the previous analysis, which was restricted to the NPS as a neuromarker of nociceptive processing, this manuscript now focuses on voxel-wise brain-activity. This allows us to investigate placebo effects on individual regions and the distribution of effects across the brain, including in regions associated with affective and cognitive processes beyond nociception.

**Data acquisition**. As previously described[7], we performed a systematic literature search to identify experimental functional magnetic resonance imaging (fMRI) investigations of placebo analgesia (see Supplementary Fig. 1, Supplementary Table 1, and ref. [7] for details). Criteria for study eligibility were: (a) published in peer-reviewed journal in the English language; (b) original investigation; (c) human participants; (d) functional neuroimaging of the brain during evoked pain; (e) pain delivered under matched placebo and control conditions. Definitions of placebo and control conditions (see Supplementary Methods and Results) were identical to our previous meta-analysis[7]. Investigators of eligible studies were contacted and invited to share data. We collected single-participant, first-level, whole-brain standard-space summary images of pain response (statistical parametric maps) from the original analyses, as published, as well as corresponding pain ratings, experimental design parameters, and demographic data (Supplementary Tables 2–5).

**Outcome definition and comparisons**. The main outcome was pain-related change in fMRI signal (i.e., blood oxygen level-dependent signal, perfusion changes), i.e., the effect of painful stimulation compared to baseline, as estimated in the original analyses (i.e., beta or contrast images). Based on this outcome, we performed three comparisons: (i) main effect of pain vs baseline, averaging placebo and control conditions; (ii) pain-related activity acquired under matched placebo and control conditions (placebo–control); and (iii), for studies that manipulated placebo vs. control within-subject, correlations across individuals between the effect of placebo treatment on brain activity and behavioral placebo analgesia (i.e., [placebo–control] in pain ratings).

Non-painful or mildly painful[12,48–52] stimulus conditions were excluded. For studies that involved left- and right-lateralized stimulation[53], strong and weak placebo conditions[54,55], or early- and late heat-pain periods[55–57] maps were averaged on subject level, as in the previous analysis[7] (see Supplementary Table 6 for details). The main effect of pain vs baseline was averaged across placebo and control conditions (instead of just using no-placebo conditions) because for some studies[48] only pooled estimates of the main effect of pain were available (see Supplementary Methods and Results for details).

**Analysis**. We applied the Cochrane risk of bias tool[58] to assess the risk-of-bias of included studies. (Supplementary Methods and Results and Supplementary Table 7).

Images underwent systematic quality control, as described previously[7] (see Supplementary Methods and Results for details). Voxels missing in >10% of participants ($n > 60$) or outside of the MNI152 brain-template (as implemented in SPM12[59], probability of brain tissue <50%) were excluded from analysis.

Outcome assessment was performed in a mass-univariate fashion, separately for each brain voxel. To account for between-study differences in the scaling of pain reports and imaging data, we used standardized effect sizes rather than the raw values. Standardization of effect sizes (mean response to pain and its difference between placebo and control conditions) was based on the between-subject-level standard deviation of pain-related brain activity, separately for each study, by using Hedges' g (Hedges' $g_{rm}$ for within-subject studies[60]), a small-sample bias-corrected version of Cohen's $d$, commonly used in meta-analysis[58]. Furthermore, we used Pearson's r to assess correlations between placebo analgesia and its effects on brain activity in studies with a within-subject design (18 studies, 460 participants). Study-level effect size estimates were summarized using the generic inverse-variance (GIV) weighting method, accounting for study as a random effect[61,62]. Pearson's r was transformed to Fisher's Z for across-study averages and tests of statistical inference. Between-study heterogeneity in effect size was estimated using the $\tau$-statistic, which represents the study-level standard deviation of effect sizes[62]. Effect size summaries and standard errors were used to calculate z-[62] and pseudo-z-scores[63], the latter was based on smoothened (4 * 4 * 4 mm full-width-half-maximum Gaussian kernel) instead of raw standard errors, as described in ref. [63]. Voxel-wise p-values were obtained by performing a non-parametric permutation-test of the pseudo-z statistic[63], correcting for multiple comparisons at Family Wise Error (FWER) level, according to the maximum-z method[63]. Permutation testing was performed at 1500 random permutations, small ($p < 0.01$) p-values were approximated by tail-fitting a generalized Pareto distribution[64]. To perform a robust enhancement of spatially extended activations and, at the same time, allowing for simple z-score-based thresholding, we performed probabilistic threshold-free cluster enhancement (pTFCE)[65]. Both enhanced and unenhanced z-score maps were thresholded at an FWER-corrected alpha level of $p < 0.05$. For visualization, unenhanced z-score maps were thresholded at an uncorrected alpha level of $p < 0.05$. All p-values presented are two-tailed.

To aid the interpretation of results, we utilized cosine similarity in an exploratory analysis comparing the participant-level contrast maps with brain-parcellation atlases representing (i) canonical large-scale functional connectivity networks[14] (resting-state), as well as (ii) insular sub-regions (anatomy based)[66], and (iii) thalamic nuclei (anatomy based)[67], i.e., the most prominent brain regions involved in pain processing[13]. Note that the large-scale functional connectivity networks are based on resting-state data, thus reverse-inference and direct associations to task-based activity should be performed carefully. On the other hand, large-scale resting-state networks have de-facto evolved as standard means for brain-wide localization[68]. Accordingly, we use these canonical networks solely for localization purposes. Obtained participant-level cosine similarity values were

summarized using the GIV method, with statistics based on *t*-tests across studies, treating study as a random effect. No correction for multiple comparisons was performed for the atlas-based analyses due to their exploratory nature. Cosine similarity is equivalent to Pearson's correlation except for mean-centering, so it remains sensitive to the overall level of activation across the brain and thus reflects absolute normalized activity levels in the regions/networks tested rather than relative activity across regions.

All analyses were performed with MATLAB 2016b, SPM12, the CANlab Core Tools neuroimaging analysis toolbox (https://github.com/canlab/CanlabCore), and custom functions implementing the GIV method. Further analysis details are provided in the Supplementary Methods and Results.

MRIcroGL (v28.5.2017) was used to create illustrations of statistical parametric maps. All neuroimages shown follow the neurological convention (left side corresponds to left hemisphere in coronal- and axial-sections). Effect sizes are interpreted as small, moderate, and large according to the recommendations by Cohen[69]. All result maps from this meta-analysis are available for download as 3d NIFTI images at https://osf.io/n9mb3/.

**Reporting summary**. Further information on research design is available in the Nature Research Reporting Summary linked to this article.

## Data availability

Results as 3d-volumes are provided at https://osf.io/n9mb3/. Participant-level source data are available from the authors upon reasonable request and with permission of the Placebo Imaging Consortium.

## Code availability

The full analysis code is available at https://github.com/mzunhammer/PlaceboImagingMetaAnalysis.

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

## Acknowledgements

This research was funded by the Mercator Research Center Ruhr (MERCUR) and National Institutes of Health R01 MH076136 (T.D.W.) and the German Research Foundation, TRR 289 Treatment Expectation. Gefördert durch die Deutsche Forschungsgemeinschaft (DFG) – Projektnummer 422744262 – TRR 289.

## Author contributions

U.B. and T.D.W. had full access to all the data in the study and take responsibility for the integrity of the data and the accuracy of the data analysis. Concept and design: M.Z., U.B., and T.D.W. Acquisition of the data: All authors. Drafting of the manuscript: M.Z., U.B., and T.D.W. Critical revision of the manuscript for important intellectual content: All authors. Statistical analysis: M.Z., T.D.W., and T.S. Visualization of Results: T.S., T.D.W., and M.Z. Obtained funding: U.B. and T.D.W. Administrative, technical, or material support: U.B. Supervision: U.B. and T.D.W.

## Funding

## Competing interests

M.Z. is a full-time employee of Takeda Pharma; the present publication has been prepared independently and outside of the employment; the employer is not involved in any of the subjects dealt within this publication and did not provide any form of support. The authors declare no competing interests.

## Additional information

## The Placebo Imaging Consortium

Lauren Atlas[3,4,5], Fabrizio Benedetti[6,7], Ulrike Bingel[1✉], Christian Büchel[8], Jae Chan Choi[9,10], Luana Colloca[11], Davide Duzzi[12], Falk Eippert[13], Dan-Mikael Ellingsen[14,15], Sigrid Elsenbruch[16], Stephan Geuter[17], Ted J. Kaptchuk[18], Simon S. Kessner[19], Irving Kirsch[18], Jian Kong[20], Claus Lamm[21], Siri Leknes[22,23], Fausta Lui[12],

Alexa Müllner-Huber[21], Carlo A. Porro[12], Markus Rütgen[21], Lieven A. Schenk[8], Julia Schmid[24], Tamás Spisák[1], Nina Theysohn[25], Irene Tracey[26], Tor D. Wager [2✉], Nathalie Wrobel[27], Fadel Zeidan[28] & Matthias Zunhammer [1]

[3]National Center for Complementary and Integrative Health, National Institutes of Health, Bethesda, MD, USA. [4]National Institute on Drug Abuse, National Institutes of Health, Baltimore, MD, USA. [5]National Institute of Mental Health, National Institutes of Health, Bethesda, MD, USA. [6]University of Turin, Turin, Italy. [7]Plateau Rosà Labs, Plateau Rosà, Switzerland. [8]Dept. of Systems Neuroscience, University Medical Center Hamburg-Eppendorf, Hamburg, Germany. [9]Yonsei University, Wonju College of Medicine, Wonju, South Korea. [10]Cham Brain Health Institute, Seoul, South Korea. [11]University of Maryland, Baltimore, MD, USA. [12]Dept. of Biomedical, Metabolic and Neural Sciences, University of Modena and Reggio Emilia, Modena, Italy. [13]Max Planck Institute for Human Cognitive and Brain Sciences, Leipzig, Germany. [14]Norwegian Center for Mental Disorders Research (NORMENT), Oslo University Hospital, Oslo, Norway. [15]Dept. of Psychology, University of Oslo, Oslo, Norway. [16]Dept. of Medical Psychology and Medical Sociology, Ruhr University Bochum, Bochum, Germany. [17]Johns Hopkins University, Baltimore, MD, USA. [18]Beth Israel Deaconess Medical, Harvard Medical School, Boston, MA, USA. [19]Dept. of Neurology, University Medical Center Hamburg-Eppendorf, Hamburg, Germany. [20]Massachusetts General Hospital, Harvard Medical School, Cambridge, MA, USA. [21]Social, Cognitive and Affective Neuroscience Unit, Dept. of Cognition, Emotion, and Methods in Psychology, Faculty of Psychology, University of Vienna, Vienna, Austria. [22]Dept. of Psychology, University of Oslo, Oslo, Norway. [23]Dept. Diagnostic Physics, Oslo University Hospital, Oslo, Norway. [24]Institute of Medical Psychology and Behavioral Immunobiology, University Hospital Essen, Essen, Germany. [25]Insitute for Diagnostic and Interventional Radiology and Neuroradiology, University Hospital Essen, Essen, Germany. [26]University of Oxford, Oxford, UK. [27]Karolinska Institute, Solna, Sweden. [28]Wake Forest School of Medicine, Winston-Salem, NC, USA.

