## [Peer Review File · Nature Communications]

Reviewers' comments:

Reviewer #1 (Remarks to the Author):

This is a well written manuscript that has a clear and straight-forward design and interpretation. The value of conducting a large 'mega-analysis' in 603 participants is well-stated and particularly relevant for the topic of placebo analgesia, where many small-scale studies have reported heterogeneous results. It provides a comprehensive go-to reference that carefully describes the effects of placebo analgesia.

The data set is well organised and carefully processed, and has previously been used to investigate the impact of placebo on the Neurologic Pain Signature. Some practical decisions were made to standardise and combine the data across studies. These appear to be sensible and robust.

The results are well-presented and highlight important new insights relating to the interaction between pain and placebo effects. These results are discussed carefully in the context of what is already known and the limitations of the study. The challenges associated with the well-understood reverse inference problem are considered in the methods and appropriately discussed.

Although, as would be expected, many of the observations reported in this study have been previously described in the original papers, there is still great value in identifying robust changes which are consistently identified across studies in this large-scale analysis, which are consistent across a variety of experimental designs. This type of 'mega-analysis' approach to brain imaging studies is still relatively novel and the Supplementary Appendix, provides an excellent example of how these complex studies should be conducted.

Rebecca Slater

Reviewer #2 (Remarks to the Author):

In this study the authors conducted a large scale meta-analysis across neuroimaging studies that collected brain data during control and placebo conditions to determine placebo effects on responses to painful stimuli. The meta-analysis provides results for pain-related effects on brain activity and placebo-related effects on brain activity. The major claims of the paper are consistent with previous meta-analyses and individual analyses in that placebo effects are largely indicated as reduced pain-related brain activity (correlated with reduced behavioral pain response during placebo). Small regional increases in brain activity are also identified by this study and indicate top-down modulation. Between-study heterogeneity was minimal and did not overlap significantly with the regions of pain-related or placebo-related effects. The authors claim that the regions identified may provide promising new targets for future studies of placebo analgesia brain mechanisms. Novel reports of cerebellar effects are reported and discussed in the paper, while these were not reported in previous meta-analyses. Overall the authors state that their results suggest placebo effects are not restricted to sensory/nociceptive or cognitive/affective processes, but may involve combined mechanism that differ based on paradigms and individual differences.

Major Concerns

1. The placebo effects largely show (and correlate with) decreases in pain-related brain response. The more interesting regions of increase during placebo (e.g., top-down modulatory mechanisms) show higher between-study heterogeneity which the authors interpret as due to variations in placebo induction type/methods across studies. The details and breakdown of placebo effects depending on these variations seems to be of key importance to providing informative results regarding placebo effects in this paper. As such, the paper would be much improved by an attempted supplementary analysis comparing placebo effects based on different methods. For

example, it appears from the table that this should at least be attempted by comparing across grouped placebo induction conditions (e.g., sug vs. sug+cond). Even if this comparison provides only uncorrected results it could be highly informative, especially to enhance the discussion/comparison to previous literature on pages 18-19. Additional comparisons could include treatment type (e.g., topical cream vs. other), sex differences, pain stimulus (e.g., heat vs. other).

2. For the pain stimulation effects part of the analysis, the authors pool conditions of pain-control and pain-placebo. It is assumed that this was done to essentially double the amount of data per subject (except for the between-subject studies) and increase power. However, this muddles the results for the pain stimulation effects because the differences in placebo vs. control conditions are included in the pain-related results. An explanation for why this approach was chosen should be provided. A supplementary analysis and results for pain stimulation effects using only control conditions (excluding placebo+pain conditions) vs. baseline should be conducted and provided for comparison.
3. The pain stimulation effects (and placebo effects) include brain regions that are outside of (and within) the regions included in the previously validated neurologic pain signature (NPS). This is interpreted by the authors that these regions therefore "show promise for explaining placebo analgesia". However, this discrepancy could be due to differences in the pain signature in this population (N=603) vs. the NPS population (N=114, from previous publication), especially because the previous publication (i.e., JAMA Neurology article included in the review materials) already showed that placebo effects on pain in these studies minimally overlapped with the validated NPS. To address this concern, it would be ideal for the authors to generate a de novo NPS on this population (N=603, using control vs baseline only, in line with comment #2 above) to compare to the validated NPS and to delineate on the figures (e.g., in a different/hashed outline, or in supplemental figures). In this way, the authors will be able to differentiate between placebo effects on pain vs. pain-response that is specific to this population.
4. The online pre-registered analysis plan does not include description of any of the goals or analyses for this manuscript. The pre-registered analysis plan is very specific to the previous JAMA Neurology publication (i.e., evaluating the NPS with regard to placebo effects). If the authors are only citing this analysis plan for specific sections such as how they arrived at the final data set – which is already in the eFigures (and updated), then this should be clarified/specified in the text. Otherwise it is misleading to the reader to believe that all of the analyses included in the present manuscript were pre-registered (and not exploratory).

Minor Concerns

1. "Mega"-analysis is used in the background sections while meta-analysis is used everywhere else. It is unclear in the text if this is an error or if the terms are being used differently in the background on purpose. Please correct or clarify the distinctions in the text.
2. The between subject design is not adequately described. It is assumed that this means that subjects did not undergo each condition (placebo and control) in these studies, but this should be stated explicitly either in the text or table legend.
3. P. 9/30 the result maps are not available for download on the specified site.
4. Figure 4 text within the thalamus figure is too small and unreadable as printed. Please increase the size of this (and other) key maps so that the details are legible.
5. Figure 4 insula subdivisions/colors are very difficult to distinguish as shown. Within the circle, the colored maps with the whole brain sagittal backgrounds are helpful. But, it is unclear which colors pertain to left vs. right insula regions (or if bilateral) and the colors are not easy to distinguish (multiple slightly different shades of green, and similar other color issues). The structural image underlays on the small insula insert maps (i.e., the outermost circle of images) unfortunately do not show enough structural topology to add any useful information for the reader so it is suggested that these be enlarged and/or "zoomed out" to show more axial/horizontal topology detail.
6. In the last sentence in the first paragraph of the discussion, it is assumed that "these regions" refers to the regions outside the NPS, but this should be clarified because it is confusing as

written.

7. The figure legends should specify that "Heterogeneity" maps refer to "between-study heterogeneity".

8. The circular graphs in Figure 4 for cosine similarity estimates do not describe that results are shown in red/orange or blue for positive or negative results, respectively (this is listed only in an eFigure), and this should be stated in the figure legend.

Reviewer #3 (Remarks to the Author):

What are the major claims of the paper?

This is a clear and well-written manuscript that aims to elucidate the neural underpinnings of placebo analgesia using a mega analysis approach. A major claim of the paper is that placebo analgesia (based on experimental pain experiments in healthy volunteers) affects pain-related activity in brain areas within and outside "typical nociceptive regions".

Are they novel and will they be of interest to others in the community and the wider field?

The same consortium recently published a paper (Zunhammer et al. JAMA Neurology 2018) where the exact same data was used to test effects within the NPS, i.e. in pain-relevant brain regions. The novelty of this paper, no matter how well executed it is, is therefore not clear (at least not the way the manuscript stands now). The authors look for placebo-related brain activations both within and outside pain-relevant brain regions. This time a priori pain regions are also considered, but not defined by the NPS. Why have the authors chosen to publish two separate papers on highly overlapping research questions using the same dataset? The authors write: "A previous paper on this dataset 7 tested placebo effects on a single, a-priori pain-related measure (the Neurologic Pain Signature 12). Here, we used the same pre-registered analysis plan (<https://osf.io/n9mb3/>) to map placebo responses across the brain." It is unclear (1) what part of the preregistration relates to this paper. (2) Why two papers are published separately using the same preregistration.

From the list of studies included in the analysis, it seems that the authors have chosen the studies carefully to reflect placebo treatment, and not a mixed bag of cognitive modulation of pain with no actual administration of treatment. This reflects authors' deep understanding of the studies included in the analysis, and hopefully adds to the validity of pooling data this way. The careful selection likely minimised unwanted heterogeneity.

Is the work convincing, and if not, what further evidence would be required to strengthen the conclusions?

The study design, and statistical approach, is convincing. For comments and detailed questions about the methods see the last section of this review.

The way the results are discussed now, they seem somewhat contradictory to the previous paper. The 2018 paper found very small effects of placebo in nociceptive brain regions, indicating that placebo affect pain via mechanisms "largely independent of nociceptive processing". Here, in the present paper, the authors state that: "In this systematic meta-analysis of individual participant data, we show that placebo treatments induce small, yet robust, inhibitory effects in large parts of the brain. These involve the ventral attention and somatomotor networks, including targets of spinothalamic afferents strongly linked to nociception, and are consistent across studies." The authors should comment on this, and reconcile the the findings in the two papers.

On a more subjective note, do you feel that the paper will influence thinking in the field?

This paper provides interesting results for anyone who is interested in the neural correlates of placebo analgesia. The paper influences the thinking around placebo and pain by showing that decreased perception of pain is not simply reflected in decreased nociceptive activations. Other brain networks interact with nociceptive regions and shape the subjective perception of pain. A very important lesson.

Method specific comments and questions

Points 1-3 raise the question if it is appropriate to perform a mega analysis on this dataset considering the amounts of heterogeneity that is not controlled for:

1) The data used in this analysis has vastly different preprocessing routines (eTable 7) from SPM99 to SPM12 to FSL (unspecified versions), something which is known to change outcomes (see e.g. Bowring et al 2019, Botvinik-Nezer et al 2019). It is unclear why, when requesting the raw data, the authors decided to request vastly heterogeneous preprocessed data with varying and outdated preprocessing routines instead of obtaining the raw data and applying the same modern preprocessing routine to it. This limitation should be stated in the limitations section. However, when using different preprocessed data there are differences which could have been adjusted/controlled for which have not (see below).

2) Smoothing. The smoothing that has been applied varies between studies (between 4-9mm). Then the analyses here place an additional 4mm smoothing here onto the results, which increases the effective smoothing even further. This appears problematic given the aim of the paper to identify brain regions relating to pain/placebo. Equalizing the smoothness of all the data first (e.g. using AFNI's 3dBlurToFWHM) would be an alternative.

3) Different MNI templates. The different preprocessing software used use different MNI templates (e.g. SPM12 FSLs use the nonlinear one from 2006 and earlier version of SPM use the linear one. Information about different MNI spaces can be read at <https://www.lead-dbs.org/about-the-mni-spaces/>). While these visually look similar, there are variations with how the functional data has been transformed onto the template. Thus phrases like "All images are aligned to the standard MNI template." (Figure 1, caption) should be avoided. Second, this could also effect some of the results and transforming all results to the same MNI template would be beneficial.

Point 4-5 comments on Figures

4) In figure 4, it is unclear why the authors have decided to show uncorrected results when it appears. Further, when reporting the p-values in the results section discussing Figure 4 the authors do not say these are uncorrected statistical tests but use comments such as "These did not reach significance" (p.g. 14).

5) Figure 4 is confusing and can be improved. First, the legend refers to 3 columns but the figure contains 4 equally sized columns (and the reader is to assume the first column should be ignored?). Second, there are no values on the plot to understand the magnitudes of the different variables (apart from the outside of the circle). Third, the amplitude of the values in the increase in both radius and circumference which is effectively comparing different circle areas which is a known cognitive bias in data visualization (An example. col "3", the dAttn is about 50% lower SomMot (I presume, no ticks!) but visually it looks considerably smaller than 50%, the impression is about 75% smaller). Forth, it is unclear if the mean r-values in column "3" have been averaged as Z-values or as r-values. Fifth, there is no label indicating whether red or blue is positive/negative in the figure. Violin, polar with points and error bars with SEM, or even bar plots would be depict the data more accurately.

References

Bowring, A., Maumet, C., & Nichols, T. E. (2019). Exploring the impact of analysis software on task fMRI results. *Human brain mapping, 40*(11), 3362-3384.

Botvinik-Nezer, R., Holzmeister, F., Camerer, C. F., Dreber, A., Huber, J., Johannesson, M., ... & Avesani, P. (2019). Variability in the analysis of a single neuroimaging dataset by many teams. *bioRxiv*.

Reviewer #1 (Remarks to the Author):

R1.0: This is a well written manuscript that has a clear and straight-forward design and interpretation. The value of conducting a large 'mega-analysis' in 603 participants is well-stated and particularly relevant for the topic of placebo analgesia, where many small-scale studies have reported heterogeneous results. It provides a comprehensive go-to reference that carefully describes the effects of placebo analgesia.

The data set is well organised and carefully processed, and has previously been used to investigate the impact of placebo on the Neurologic Pain Signature. Some practical decisions were made to standardise and combine the data across studies. These appear to be sensible and robust.

The results are well-presented and highlight important new insights relating to the interaction between pain and placebo effects. These results are discussed carefully in the context of what is already known and the limitations of the study. The challenges associated with the well-understood reverse inference problem are considered in the methods and appropriately discussed.

Although, as would be expected, many of the observations reported in this study have been previously been described in the original papers, there is still great value in identifying robust changes which are consistently identified across studies in this large-scale analysis, which are consistent across a variety of experimental designs. This type of 'mega-analysis' approach to brain imaging studies is still relatively novel and the Supplementary Appendix, provides an excellent example of how these complex studies should be conducted.

Rebecca Slater

Author's Response R1.0:

Dear Prof. Slater, we would like to thank you for the concise evaluation of our manuscript and for highlighting the relevance and strengths of our study.

Reviewer #2 (Remarks to the Author):

R 2.0 In this study the authors conducted a large scale meta-analysis across neuroimaging studies that collected brain data during control and placebo conditions to determine placebo effects on responses to painful stimuli. The meta-analysis provides results for pain-related effects on brain activity and placebo-related effects on brain activity. The major claims of the paper are consistent with previous meta-analyses and individual analyses in that placebo effects are largely indicated as reduced pain-related brain activity (correlated with reduced behavioral pain response during placebo). Small regional increases in brain activity are also identified by this study and indicate top-down modulation. Between-study heterogeneity was minimal and did not overlap significantly with the regions of pain-related or placebo-related effects. The authors claim that the regions identified may provide promising new targets for future studies of placebo analgesia brain mechanisms. Novel reports of cerebellar effects are reported and discussed in the paper, while these were not reported in previous meta-analyses. Overall the authors state that their results suggest placebo effects are not restricted to sensory/nociceptive or cognitive/affective processes, but may involve combined mechanism that differ based on paradigms and individual differences.

Author's Response R2.0: We'd like to thank Reviewer #2 for the thoughtful and constructive review, especially for pointing out opportunities to further improve the manuscript and for highlighting additional research directions. We have addressed these helpful suggestions, which included additional data analyses and revised our manuscript accordingly. We hope our corrections have satisfactorily addressed the points raised below.

Major Concerns

R2.1. The placebo effects largely show (and correlate with) decreases in pain-related brain response. The more interesting regions of increase during placebo (e.g., top-down modulatory mechanisms) show higher between-study heterogeneity which the authors interpret as due to variations in placebo induction type/methods across studies. The details and breakdown of placebo effects depending on these variations seems to be of key importance to providing informative results regarding placebo effects in this paper. As such, the paper would be much improved by an attempted supplementary analysis comparing placebo effects based on different methods. For example, it appears from the table that this should at least be attempted by comparing across grouped placebo induction conditions (e.g., sug vs. sug+cond). Even if this comparison provides only uncorrected results it could be highly informative, especially to enhance the discussion/comparison to previous literature on pages 18-19. Additional comparisons could include treatment type (e.g., topical cream vs. other), sex differences, pain stimulus (e.g., heat vs. other).

Author's Response R2.1:

We fully agree that the research directions highlighted by the reviewer are of high relevance and interest for pain and placebo research and beyond. It has long been discussed that there is not “one placebo effect, but many”, and we are committed to the long-term goal of dissecting their underlying CNS mechanisms, as is evident by our previous work on pain modulatory mechanisms^{1,2}.

Based on the reviewer's suggestion, we have now included an exploratory sub-group comparison of the “placebo vs non-placebo control” contrast for “conditioning + suggestion” vs “suggestion only” studies as Supplementary eFigure 10.

As can be seen there are merely a few regions in the parietal white matter that surpass $p < .001$ (uncorr), the more heterogenous regions of the pre-frontal cortex mentioned above do not show any significant voxels or a clear pattern in this analysis.

These interesting yet preliminary findings have to be seen in the light of several major methodological considerations/constraints:

- Comparisons between different varieties of pain, placebo, sample, or imaging types are necessarily comparisons between sub-groups of studies, not between, or within participants. This limits the sensitivity of such analyses. Study is the unit of analysis (random effect) for such comparisons, and with 20 studies we are likely to be under-powered.
- Further, the 20 studies involved in this analysis cluster in terms of placebo-, pain- and imaging-related features. A simple between-study group comparison can thus be confounded by correlated properties across studies. For example, the type of conditioning (conditioning+suggestion vs. suggestion alone) is not balanced with respect to the stimulus modality (heat, mechanical, visceral, laser, electrical), so effects related to suggestion could be masked or artificially enhanced by virtue of correlations with other variables.

The same challenges apply to other analyses suggested by the reviewer, e.g. regarding the type of painful stimulation or placebo (topical vs. systemic).

One approach to this would be meta-regression, which considers these between-study characteristics while controlling for others. However, this analysis is based on a fundamentally different methodology (hierarchical meta-regression, not GIV), and the methods are not fully implemented yet for this type of mixed-effects data. We have started working on this approach on whole-brain level, but considering that the methods have not yet been implemented and the ability to obtain valid results in a limited sample size of 20 studies remains unclear, we consider this type of analysis to be out of the scope of the present paper. We are currently planning to publish meta-regression results in a subsequent paper if these issues can be

satisfactorily resolved, and have included a note about this in the Discussion (p.19: “Understanding the neural and neurochemical pathways underlying this variability will pave the way to systematically utilize/modulate placebo responses in a context-, patient-, and disease-specific manner.”).

R2.2 For the pain stimulation effects part of the analysis, the authors pool conditions of pain-control and pain-placebo. It is assumed that this was done to essentially double the amount of data per subject (except for the between-subject studies) and increase power. However, this muddles the results for the pain stimulation effects because the differences in placebo vs. control conditions are included in the pain-related results. An explanation for why this approach was chosen should be provided. A supplementary analysis and results for pain stimulation effects using only control conditions (excluding placebo+pain conditions) vs. baseline should be conducted and provided for comparison.

Author's Response R2.2:

We gladly adopt the reviewer's suggestion to add the pain vs baseline contrast for the control condition to the supplement (see: new Supplementary eFigure 6 and to the set of shared result volumes in .nii format, which are now attached for review). Certainly, the latter will be of particular interest for the broader field of pain-research (e.g. for defining ROIs). A comparison of baseline-only compared to the pooled analysis shows mostly the same brain regions after FWE-thresholding, yet – as expected given the smaller power, blobs tend to be smaller in the “control only” sample (in particular de-activation). As the reviewer states, this is likely due to the lower statistical power.

We would like to clarify that we pooled pain across placebo conditions for several reasons. First, as stated in the Methods section p.7 “[...]because for some studies³ only pooled estimates of the main effect of pain were available.” Showing “control condition only” would not represent the complete sample on which meta-analysis is based. Secondly, defining ROIs in the Control condition only and then testing it on Placebo vs. Control can produce biased results due to regression to the mean, as the voxel selection conditions are not independent of the test contrast. Such biases are discussed at length in Kriegeskorte et al. 2009⁴ and in our own work (e.g., *Principles of fMRI* by Lindquist and Wager, and *Fundamentals of Functional Neuroimaging* Geuter et al. 2014). A commonly used solution is to define ROIs using a contrast that is orthogonal to the test contrast of interest—e.g., for conditions A and B, defining ROIs based on A+B and testing on A-B. That is what we do here, defining ROIs on [Placebo + Control] and comparing these “pain-related” regions to effects of [Placebo – Control]. This is the approach advocated in Kriegeskorte et al. 2009⁴ and Friston et al. 2006⁵. If we defined regions (or an NPS-like pattern, as suggested below) based on the Control condition only, the Placebo condition would be less responsive even in the absence of true Control–Placebo effects due to regression to the mean.

For the sake of brevity, we did not extend the justification in the main text, but included the following extended explanation in the supplementary methods:

“For the pain vs baseline comparison we pooled placebo & control conditions based on four considerations:

- I) for some studies³ only pooled estimates of the main effect of pain were available, the map based on “control images only” would not show the complete sample*
- II) pooling provides a map that is optimal for comparisons between the pain and the placebo contrast, as the two contrasts are orthogonal. (Comparisons based on the baseline-contrast only would be biased towards the control condition)*
- III) pooling reduces within-subject variance and therefore robustness of results*
- IV) the range of effect sizes observed for the pain vs baseline comparison was about 7 times greater than that observed for the placebo vs control comparison, so placebo-related effects do not affect the visualization of the pain vs baseline comparison at large.*

R2.3. The pain stimulation effects (and placebo effects) include brain regions that are outside of (and within) the regions included in the previously validated neurologic pain signature (NPS). This is interpreted by the authors that these regions therefore “show promise for explaining placebo analgesia”. However, this discrepancy could be due to differences in the pain signature in this population (N=603) vs. the NPS population (N=114, from previous publication), especially because the previous publication (i.e., JAMA Neurology article included in the review materials) already showed that placebo effects on pain in these studies minimally overlapped with the validated NPS. To address this concern, it would be ideal for the authors to generate a de novo NPS on this population (N=603, using control vs baseline only, in line with comment #2 above) to compare to the validated NPS and to delineate on the figures (e.g., in a different/hashed outline, or in supplemental figures). In this way, the authors will be able to differentiate between placebo effects on pain vs. pain-response that is specific to this population.

Author’s Response R2.3:

The reviewer suggests that placebo-induced changes in basic pain processing may have gone unnoticed by the NPS analysis⁶, because its constituting voxels deviate from the theoretical optimum pain signature of the sample at hand. The reviewer suggests to train a “placebo-meta-specific NPS”, based on pain > baseline contrasts), with on the sample at hand (pain > baseline contrast) and re-test the hypothesis. This is an interesting proposal, and a “placebo-meta NPS” may indeed be more sensitive to the pain>baseline within the sample at hand. However, after carefully considering this, we believe this analysis will not provide a stronger or more definitive test of placebo effects on a “pain signature” than that already provided in Zunhammer et al. 2018, for several reasons:

- Training a “placebo-meta-specific NPS” on Control images only would produce biases due to regression to the mean, as described above. Using pooled images would be a better choice, since these would a) be unbiased (see Friston et al. 2006), (b) allow to include pain>control data from all studies in the sample, c) have more statistical power.
- Training and testing a predictive pattern on the same sample comes with the danger of reducing the generalizability of results. Using a pain-predictive measure developed in a held-out/separate training data-set (as is the case for the NPS) is best practice in machine-learning.
- The pain>baseline contrast in the present sample is not specific enough for creating a valid predictor for pain-related processes, as it will likely be confounded with non-noxious somatosensory processes. A [Pain – Baseline] classifier could capture many processes, e.g., general attention or negative affect, that are not specific to pain. By contrast, the NPS was trained to track pain across 4 intensities calibrated for each person to span from just below pain threshold to very painful. More importantly, its specificity to pain and generalizability across cohorts and types of pain have been validated in over 40 published cohorts. A newly trained pattern would not compare in terms of quality or require extensive validation. For this reason, we consider the NPS a more complete test of a “pain marker”, and did not seek to reduplicate this effort here.
- Finally, our previous analysis showed that the original NPS captured the pain response quite well in this sample⁶ (detecting a positive response for > 95% of participants, with an effect size). Therefore, the NPS did appear to capture pain signaling adequately in this set of studies.

For these reasons, and the fact that the present study focuses on voxel-wise analyses of local features affected by placebo across the brain (in distinction to our previous work on multivariate pain-related measures, i.e., the NPS (please also see Response R3.3) we considered the refrain from performing the suggested analysis. If the editors and reviewers disagree and feel that this analysis is essential for the paper, we would be happy to reconsider. To further address this comment, we have added Supplementary Figure (eFigure 13) that allows for a visual comparison of the results of the present analysis and the brain regions delineating the NPS. This figure nicely illustrates that the interesting placebo-related changes identified in the present analysis only partly overlap with the NPS. Moreover, we have extended the discussion on pages 15-17 to cover this point in more detail.

R2.4: The online pre-registered analysis plan does not include description of any of the goals or analyses for this manuscript. The pre-registered analysis plan is very specific to the previous JAMA Neurology publication (i.e., evaluating the NPS with regard to placebo effects). If the authors are only citing this analysis plan for specific sections such as how they arrived at the final data set – which is already in the eFigures (and updated), then this should be clarified/specified in the text. Otherwise it is misleading to the reader to believe that all of the analyses included in the present manuscript were pre-registered (and not exploratory).

Author’s Response R2.4:

We apologize, the sentence “Here we used the [...] same pre-registered analysis plan” was not precise enough. The present analysis followed the pre-registered NPS analysis in most key aspects (data acquisition, pre-processing, definition of groups/comparisons, even GIV-functions/formulas). However, the original analysis plan solely included the NPS-analysis as a main outcome. Though we followed the same plan as the pre-registered NPS analysis plan, the voxel-wise brain-wide analysis presented here was not included in the original plan and therefore not formally pre-registered. To clarify this point, we made the following changes to the manuscript:

The reference to the analysis plan was removed from the manuscript. The sentence now mentions the data set, only:

“Here, we used the same dataset to map placebo responses across the brain.”

At the same time, we elaborate on the issue in the online supplement, by stating that:

“The present analysis was not pre-registered, yet performed according to the analysis plan for Zunhammer et al. (2018)⁷ (see: <https://osf.io/n9mb3/>), with the difference that single-voxel brain responses were the main outcome, not NPS responses. Of note, statistical thresholds were not pre-defined in the original analysis plan. Therefore we provide maps for several established thresholding methods, i.e. uncorrected at $p < .001$ (parametric p -values), family-wise error (FWE) corrected at $p < .05$ (non-parametric permutation-based p -values), with and without probabilistic threshold-free cluster enhancement⁸.”

Minor Concerns

R2.5: “Mega”-analysis is used in the background sections while meta-analysis is used everywhere else. It is unclear in the text if this is an error or if the terms are being used differently in the background on purpose. Please correct or clarify the distinctions in the text.

Author’s Response R2.5:

We agree with the Reviewer’s concern: We further reduced the use of “mega analysis” to two mentions in the introduction. We prefer to use the longer, but more precise term “participant level meta analysis” in the rest of the manuscript. Nevertheless, we still introduce the term “mega-analysis” briefly in the introduction section, as it is a common synonym of “single-participant level meta-analysis in neuroimaging”. Mentioning the term may facilitate web-based searches of the paper and be advantageous for some readers using/recognizing the term.

R2.6: The between subject design is not adequately described. It is assumed that this means that subjects did not undergo each condition (placebo and control) in these studies, but this should be stated explicitly either in the text or table legend.

Author’s Response R2.6:

The reviewer is correct, “between subject design” was meant to describe studies where subjects did not undergo each condition, but were instead randomly assigned to Placebo or Control conditions. The meta-

analysis included both within-subject and a between-group design studies (see: Table 1). For clarity, the legends of Table 1 and Supplementary eTable 3 (previously eTable 4) were updated to resolve any ambiguity regarding the terms “within” and “between”.

R2.7: P. 9/30 the result maps are not available for download on the specified site.

Author’s Response R2.7:

We did not publicly upload the images yet, but will do so at the time the manuscript is published. We have now attached a package of said brain volumes for review. In correspondence with the editor, we will upload the package to the OSF repository upon publication of the manuscript.

R2.8: Figure 4 text within the thalamus figure is too small and unreadable as printed. Please increase the size of this (and other) key maps so that the details are legible.

Author’s Response R2.8:

Figure 4 has been revised thoroughly and the size of labels has been increased. Please note that Figure 4 consists of several sub-charts that we expect to re-format for publication upon requests of the editorial team.

R2.9: Figure 4 insula subdivisions/colors are very difficult to distinguish as shown. Within the circle, the colored maps with the whole brain sagittal backgrounds are helpful. But, it is unclear which colors pertain to left vs. right insula regions (or if bilateral) and the colors are not easy to distinguish (multiple slightly different shades of green, and similar other color issues). The structural image underlays on the small insula insert maps (i.e., the outermost circle of images) unfortunately do not show enough structural topology to add any useful information for the reader so it is suggested that these be enlarged and/or “zoomed out” to show more axial/horizontal topology detail.

Author’s Response R2.9:

This analysis is based on established atlases that are described in detail in the original publication. The coloring for the insular sub-regions refers to the regions bilaterally. Though space is necessarily limited in the main figures, we have added Supplementary Figures showing enlarged regions with more detail.

R2.10: In the last sentence in the first paragraph of the discussion, it is assumed that “these regions” refers to the regions outside the NPS, but this should be clarified because it is confusing as written.

Author’s Response R2.10:

Thank you for bringing this to our attention. The two sentences have been re-arranged and shortened to eliminate ambiguity:

“The NPS analysis did not provide a broader test of regions affected by placebo treatment and showed only very small effects of placebo analgesia. The present results are important because they show changes outside of the NPS, in frontal-parietal regions, in the habenula, and in specific parts of the insula and thalamus, which are promising targets for explaining placebo analgesia.”

R2.11. The figure legends should specify that “Heterogeneity” maps refer to “between-study heterogeneity”.

Author’s Response R2.11:

Thank you for bringing this to our attention. We have amended the figure legends (and a couple of mentions in the main text) accordingly.

R2.11: The circular graphs in Figure 4 for cosine similarity estimates do not describe that results are shown in red/orange or blue for positive or negative results, respectively (this is listed only in an eFigure), and this should be stated in the figure legend.

Author’s Response R2.11:

Again, thank you for bringing this to our attention. We have amended legends for Figure 4 accordingly.

Reviewer #3 (Remarks to the Author):

What are the major claims of the paper?

This is a clear and well-written manuscript that aims to elucidate the neural underpinnings of placebo analgesia using a mega analysis approach. A major claim of the paper is that placebo analgesia (based on experimental pain experiments in healthy volunteers) affects pain-related activity in brain areas within and outside “typical nociceptive regions”.

Author’s Response R3.0:

We thank Reviewer #3 for his thorough review of our meta-analysis, and are grateful for the constructive suggestions. We hope we have managed to improve our manuscript adequately.

R3.1 Are they novel and will they be of interest to others in the community and the wider field?

The same consortium recently published a paper (Zunhammer et al. JAMA Neurology 2018) where the exact same data was used to test effects within the NPS, i.e. in pain-relevant brain regions. The novelty of this paper, no matter how well executed it is, is therefore not clear (at least not the way the manuscript stands now). The authors look for placebo-related brain activations both within and outside pain-relevant brain regions. This time a priori pain regions are also considered, but not defined by the NPS. Why have the authors chosen to publish two separate papers on highly overlapping research questions using the same dataset?

Author's Response R3.1:

We would like to thank the reviewer for giving us the opportunity to clarify the rationale and relevance of the two different methodological approaches, and the scopes of this and our previous⁶ manuscript. As with many large datasets, it is impractical to include all relevant analysis approaches and results in a single manuscript, for at least three reasons: (1) Care and attention must be taken to adequately describe even a single analysis approach, and a full treatment of either the NPS response approach or voxel-wise approach requires a full manuscript-length description to adequately characterize the relevant background, methods, and results; (2) Mixing findings from different analysis approaches can become confusing for readers and dilute the clarity of the findings; (3) the approaches we took yielded different types of findings that, we believe, merit their own discussion and conclusions in a dedicated paper.

The first publication⁶ solely focused on the question whether placebo analgesia involves changes in the “Neurological Pain Signature”⁹. The NPS is a machine-learning-based, weighted, multi-voxel summary metric that can be interpreted as a neuromarker of nociceptive processing. In other words, the NPS can tell us “whether and to what extent does placebo analgesia affect nociceptive processing”. The NPS mainly owes its sensitivity to the machine-learning-based voxel weighting, not the localization of voxels. The NPS can primarily only be interpreted “as a whole”, because the validations of sensitivity and specificity compared with various other affective states is made on the signature response as a whole. This approach does not allow differentiated inferences on “*where something happened in the brain*”. Our two key findings in this first application were:

- (i) validation of the NPS as a neuromarker of pain (analysis of the control condition) in a large sample size and
- (ii) behavioral placebo analgesia was associated with only minimal changes in the NPS – pointing towards the relevance of other brain areas and networks for placebo analgesia. The first manuscript could only speculate on these brain areas as the analyses were solely restricted to the NPS.

The present series of analyses builds exactly upon the latter question and now focuses on single-voxel brain-activity, allowing us to answer the “where” question, “*which brain regions are affected by placebo treatment*”. *Specifically, we mapped pain- and placebo-related effects in brain activity, which revealed novel insights into brain systems involved in placebo analgesia, including the habenula and the cerebellum, both regions that were systematically missed in previous smaller-scale analyses and are not part of the NPS.*

We now further clarify the distinction in analyses in the methods, (p.:6: “*In contrast to the previous NPS-based analysis the manuscript addresses the “where do we find placebo-associated activation changes in the brain”.*”) and discussion section (p.15, reworded for clarity).

R3.2 The authors write: “A previous paper on this dataset 7 tested placebo effects on a single, a-priori pain-related measure (the Neurologic Pain Signature 12). Here, we used the same pre-registered analysis plan (<https://osf.io/n9mb3/>) to map placebo responses across the brain.” It is unclear (1) what part of the preregistration relates to this paper.

(2) Why two papers are published separately using the same preregistration.

Author’s Response R3.2:

Please see Author’s comment R2.4 for the actions taken: We removed the claim in question (p.6 “Here we used the [...] same pre-registered analysis plan” was not precise enough) and clarified the relation of the present manuscript to the pre-registration on p.8 of the Supplement.

R3.3 From the list of studies included in the analysis, it seems that the authors have chosen the studies carefully to reflect placebo treatment, and not a mixed bag of cognitive modulation of pain with no actual administration of treatment. This reflects authors’ deep understanding of the studies included in the analysis, and hopefully adds to the validity of pooling data this way. The careful selection likely minimised unwanted heterogeneity.

Is the work convincing, and if not, what further evidence would be required to strengthen the conclusions?

The study design, and statistical approach, is convincing. For comments and detailed questions about the methods see the last section of this review.

Author’s Response R3.3:

Thank you for these supportive comments.

R3.4 They way the results are discussed now, they seem somewhat contradictory to the previous paper. The 2018 paper found very small effects of placebo in nociceptive brain regions, indicating that placebo affect pain via mechanisms “largely independent of nociceptive processing”. Here, in the present paper, the authors state that: “In this systematic meta-analysis of individual participant data, we show that placebo treatments induce small, yet robust, inhibitory effects in large parts of the brain. These involve the ventral attention and somatomotor networks, including targets of spinothalamic afferents strongly linked to nociception, and are consistent across studies.” The authors should comment on this, and reconcile the findings in the two papers.

Author’s Response 3.4: This is a very astute comment, as integrating the two sets of findings requires a nuanced view of what “pain processing” is. The NPS provides one well-validated brain pattern linked to nociceptive pain – but, as we have argued in a number of publications, reflects only a neurophysiological component of pain, not all of “pain processing”. The present findings show results in some distinct areas (habenula, striatum, cerebellum) as well as areas similar (but not identical) to those involved in the NPS (somatosensory portions of the thalamus). As is typical with brain research, we must interpret these changes in light of known neuroanatomical pathways and other functional findings. Broadly speaking, the placebo-related changes we observed could be related to (a) processes related to affect, mood, and cognition that co-occur with pain but are not related to pain construction itself, or (b) alterations in pain construction processes that are not captured by the NPS. A strength of the NPS approach is its extensive validation in relation to other types of studies, but this is not possible for voxel-wise analyses. Thus, we leave the interpretation flexible for many of the findings. Some areas strongly co-localized with nociceptive afferent pathways (e.g., ventro-posterior lateral thalamus [VPL]) we interpret (with appropriate caveats) as pain construction-related. This is broadly consistent with the previous findings

that there are some significant effects of placebo on nociceptive pain representations as revealed by the NPS and significant correlations between NPS and individual differences in placebo analgesia; they are just too small to be the “whole story.” The VPL findings here and similar themselves provide a new kind of support for effects on nociceptive pain that the NPS could not. The NPS was trained on a small sample and anatomical templates that localized the VPL were not available at the time, and the approach was not designed to isolate specific pathways. Here, we used a much larger sample and newly available anatomical templates for thalamic subregions. Thus, we view the present findings in nociceptive pathways as complementing the NPS findings. In both papers, nociceptive processing appears to be part of the story of how placebo effects come about, but not the whole story. We have now extended our discussion on pages 15-17 to emphasize differences in interpretation point in more detail.

R 3.5: On a more subjective note, do you feel that the paper will influence thinking in the field?

This paper provides interesting results for anyone who is interested in the neural correlates of placebo analgesia. The paper influences the thinking around placebo and pain by showing that decreased perception of pain is not simply reflected in decreased nociceptive activations. Other brain networks interact with nociceptive regions and shape the subjective perception of pain. A very important lesson.

Author's Response R3.5:

Again, thank you for the encouraging words.

Method specific comments and questions

Points 1-3 raise the question if it is appropriate to perform a mega analysis on this dataset considering the amounts of heterogeneity that is not controlled for:

3.6 The data used in this analysis has vastly different preprocessing routines (eTable 7, now: eTable 5) from SPM99 to SPM12 to FSL (unspecified versions), something which is known to change outcomes (see e.g. Bowring et al 2019, Botvinik-Nezer et al 2019). It is unclear why, when requesting the raw data, the authors decided to request vastly heterogeneous preprocessed data with varying and outdated preprocessing routines instead of obtaining the raw data and applying the same modern preprocessing routine to it. This limitation should be stated in the limitations section. However, when using different preprocessed data there are differences which could have been adjusted/controlled for which have not (see below).

Author's Response R3.8:

We agree that the collection of raw data would have been advantageous in terms of data quality control, statistical power, and analysis opportunities (e.g. functional connectivity). We've now added a section to the Supplementary methods (p. 5) to explain why we decided to collect first-level statistical summaries, rather than raw time series-level data:

Data collection:

Investigators of all eligible studies were contacted and invited to share data. Specifically, we requested participant-level summary images (statistical parameter estimates, or beta-images) representing any relevant experimental condition. The decision to collect pre-processed, summarized participant-level images (aka 1st-level images) was based on the following considerations:

- 1. Raw image data are, unfortunately, still not typically not stored in a format that can be easily retrieved and integrated across studies. It took us approximately 1.5 years to collect person-level summary statistic maps from the original study authors, which are about 2 orders of magnitude easier to retain and share, and are standardized. It is very unlikely that we could have obtained raw image data for a reasonable subset of these studies in a reasonable time frame.*
- 2. Raw images may contain personal information (meta-data, anatomical features captured in images) that could make individual research-participants identifiable. Sharing of such images across workgroups may only be possible after consultation of local ethics committees. Additional measures (removal of meta-data and face-masking) would have to be taken to ensure participant anonymity. Meta-data of statistical summary images from SPM and fsl do not contain individual information by default and therefore safeguard anonymity.*
- 3. The analysis of neuroimaging data is an elaborate multi-step process that involves numerous analysis decisions. A multitude of opinions exist regarding the optimal analysis pipeline, especially when it comes to expressing an experimental (stimulus) protocol as a statistical model*

(most often a GLM). The “optimal” analysis depends on many considerations, some of which cannot be based on data alone. We relied on the expertise of the original researchers to choose the best approach for the data at hand.

4. *When collecting raw imaging data, the associated experimental stimulus protocols have to be collected for analysis. These often do not come in a standardized format. Re-modelling the statistical analysis in terms of pain and placebo-conditions is therefore laborious and error prone. Further, re-modelling the data requires many decisions on the side of the meta-analyst that cannot be pre-registered. This poses a potential source of “researcher degrees of freedom” and therefore bias that we wanted to avoid.*

R3.9a: Smoothing. The smoothing that has been applied varies between studies (between 4-9mm). Then the analyses here place an additional 4mm smoothing here onto the results, which increases the effective smoothing even further.

Author's Response R3.9a:

Please note that we did not apply additional smoothing to the participant-level or study-level summary images. We merely applied additional 4 mm smoothing to the *standard-error* images (s. Methods, p.8) for the purpose of obtaining pseudo-z-scores for permutation testing, in accord with Nichols & Holmes (2002)¹⁰.

R3.9b: This appears problematic given the aim of the paper to identify brain regions relating to pain/placebo. Equalizing the smoothness of all the data first (e.g. using AFNI's 3dBlurToFWHM) would be an alternative.

Author's Response R3.9b:

We've considered equalizing the smoothness in the beginning of this project, but decided to summarize the data at given smoothness. We've added the considerations for this decision to the Supplementary Methods section (p. 8):

Smoothing

The statistical summary images collected differed in terms of image smoothness (see: eTable 5). Between-study Differences in smoothing kernel may impact negatively on the comparability of single studies and the statistical weight of individual studies to the meta-analysis. However, no measures were taken to equalize image smoothness before meta-analysis based on the following considerations:

- *The main purpose of equalizing image smoothness is to achieve a better comparability of studies.¹¹ However, the present study primarily aimed at was to summarize brain activity across studies, not to make comparisons between individual studies.*
- *“One disadvantage of post hoc smoothness equalization is that it requires that all scanners be smoothed to that of the most smooth scanner in the set”¹¹. Equalizing smoothing would therefore come with a loss in statistical power and mapping-accuracy.*

Moreover, we now highlight this issue in the “limitations” section of the discussion (p. 20).

Further, the fact that this meta-analysis was based on participant-level statistical summary images from variety of different pre-processing pipelines (s. eTable 5) likely had a negative impact on spatial brain mapping accuracy, in particular since different software packages¹² (and therefore MNI-templates¹³) and different spatial smoothing kernels were used.

R3.10: Different MNI templates. The different preprocessing software used use different MNI templates (e.g. SPM12 FSLs use the nonlinear one from 2006 and earlier version of SPM use the linear one. Information about different MNI spaces can be read at <https://www.lead-dbs.org/about-the-mni-spaces/>). While these visually look similar, there are variations with how the functional data has been transformed onto the template. Thus phrases like “All images are aligned to the standard MNI template.” (Figure 1, caption) should be avoided. Second, this could also affect some of the results and transforming all results to the same MNI template would be beneficial.

Author’s Response R3.10:

We fully agree that differences in MNI templates are an issue that may add between-study heterogeneity, impact negatively on the statistical power of the analysis, and the spatial accuracy of brain mapping results, especially when compared to a de-novo analysis based on raw images. We therefore made the following changes to the manuscript:

- The misleading wording in the figure captions (“All images are aligned to the standard MNI template.”) has been removed
- We now specify the that we used the MNI152 brain-template “*as implemented in SPM12*” (*Methods, p7*).
- Additionally, we now emphasize this issue in the limitations section of the discussion. *Further, the fact that this meta-analysis was based on participant-level statistical summary images from variety of different pre-processing pipelines (s. eTable 5) likely had a negative impact on spatial brain mapping accuracy, in particular since different software packages¹² (and therefore MNI-templates¹³) and different spatial smoothing kernels were used.*

Point 4-5 comments on Figures

R3.11 In figure 4, it is unclear why the authors have decided to show uncorrected results when it appears. Further, when reporting the p-values in the results section discussing Figure 4 the authors do not say these are uncorrected statistical tests but use comments such as “These did not reach significance” (p.g. 14).

Author’s Response R3.11:

These analyses were performed to “*To aid the interpretation of results[...]*” (*Methods*, p. 8) and networks/ROIs were chosen post-hoc. We consider these analyses exploratory and did not apply correction for multiple comparisons (we are not able to account for all other possible post-hoc analyses tested).

We fully agree that an exploratory analysis of atlas-based systems/ROIs should not be discussed in a language that suggests a null-hypothesis-statistical-significance (NHST) testing framework, but in a descriptive/hypothesis-generating way. Therefore,

- we’ve further extended the description on p. 8 to flag these analyses as exploratory more clearly.
- we revised the wording of our results on pp. 13f to reduce the focus on statistical significance / p-values
- we revised our discussion on pp. 15f and now focus on a more descriptive, effect-sized based interpretation.

R3.12 a: Figure 4 is confusing and can be improved.

Author’s Response R3.12 a:

Thank you for the constructive suggestions regarding Figure 4. We have revised the figure mock-up and sub-figures, please find a point-by-point response below.

R3.12 b: First, the legend refers to 3 columns but the figure contains 4 equally sized columns (and the reader is to assume the first column should be ignored?).

Author’s Response R3.12b:

We interpreted the first column containing the atlas depictions as a row header, not a column in its own right. We’ve adapted your suggestion to reference the ROI depictions as a column, but we are open to any editorial suggestions on how to label/reference sub-parts of the figure to conform with the standards of Nature Communications.

R3.12 c: Second, there are no values on the plot to understand the magnitudes of the different variables (apart from the outside of the circle).

Author’s Response R3.12c:

We agree that this is a limitation of the radial plot design chosen and therefore we now show horizontal dot + error bar charts.

R3.12d: Third, the amplitude of the values in the increase in both radius and circumference which is effectively comparing different circle areas which is a known cognitive bias in data visualization (An example. col "3", the dAttn is about 50% lower SomMot (I presume, no ticks!) but visually it looks considerably smaller than 50%, the impression is about 75% smaller).

Author's Response R3.12d:

We agree that the cognitive biases on area perception is a limitation of the radial plots. We've switched to horizontal point + error bar charts (see below). (A side-note: Were aware that the visual focus of the original chart would be on the length *and* area of wedges. Yet in wedge plots displaying r , the area of the wedges is proportional to r^2 ... we thought this an elegant way to visualize two statistics of interest at once).

R3.12e: Forth, it is unclear if the mean r-values in column "3" have been averaged as Z-values or as r-values.

Author's Response R3.12e:

Previously, this was noted in the methods (p.8): "[...] we used Pearson's r (transformed to Fisher's Z , for inferential testing) to assess correlations [...]".

We now state this more clearly in the methods section (p.8: "*Pearson's r was transformed to Fisher's Z for all averaging and inferential testing.*"). For the sake of brevity and to avoid to confuse readers regarding the metric shown (Fisher's Z , rather than Pearson's r re-transformed), we decided not to mention Fisher's Z in the figure legends.

R3.12f: Fifth, there is no label indicating whether red or blue is positive/negative in the figure. Violin, polar with points and error bars with SEM, or even bar plots would be depict the data more accurately.

Author's Response R3.12f:

Thank you for bringing this to our attention, we've added an explanation of colors to Figure legend 4. As stated above, we opted for dot + error bar charts as a straightforward display. We opted against violin plots or plots including single data-points as it would become difficult to reflect the hierarchical (participant/study-level) nature of the data.

References

1. Woo C-W, Schmidt L, Krishnan A, et al. Quantifying cerebral contributions to pain beyond nociception. *Nat Commun*. 2017;8:14211. doi:10.1038/ncomms14211
2. Ploner M, Bingel U, Wiech K. Towards a taxonomy of pain modulations. *Trends Cogn Sci*. 2015;19(4):180-182. doi:10.1016/j.tics.2015.02.007
3. Atlas LY, Whittington R a, Lindquist M a, Wielgosz J, Sonty N, Wager TD. Dissociable influences of opiates and expectations on pain. *J Neurosci*. 2012;32(23):8053-8064. doi:10.1523/JNEUROSCI.0383-12.2012
4. Kriegeskorte N, Simmons WK, Bellgowan PS, Baker CI. Circular analysis in systems neuroscience: The dangers of double dipping. *Nat Neurosci*. 2009;12(5):535-540. doi:10.1038/nn.2303
5. Friston KJ, Rotshtein P, Geng JJ, Sterzer P, Henson RN. A critique of functional localisers. *Neuroimage*. 2006;30(4):1077-1087. doi:10.1016/j.neuroimage.2005.08.012
6. Zunhammer M, Bingel U, Wager TD, Placebo Imaging Consortium. Placebo Effects on the Neurologic Pain Signature: A Meta-analysis of Individual Participant Functional Magnetic Resonance Imaging Data. *JAMA Neurol*. 2018;75(11):1321-1330. doi:10.1001/jamaneurol.2018.2017
7. Zunhammer M, Bingel U, Wager TD. Placebo Effects on the Neurologic Pain Signature: A Meta-analysis of Individual Participant Functional Magnetic Resonance Imaging Data. *JAMA Neurol*. 2018. doi:10.1001/jamaneurol.2018.2017
8. Spisák T, Spisák Z, Zunhammer M, et al. Probabilistic TFCE: A generalized combination of cluster size and voxel intensity to increase statistical power. *Neuroimage*. 2019;185:12-26. doi:10.1016/j.neuroimage.2018.09.078
9. Wager TD, Atlas LY, Lindquist M a, Roy M, Woo C-W, Kross E. An fMRI-based neurologic signature of physical pain. *N Engl J Med*. 2013;368(15):1388-1397. doi:10.1056/NEJMoa1204471
10. Nichols TE, Holmes AP. Nonparametric permutation tests for functional neuroimaging: a primer with examples. *Hum Brain Mapp*. 2002;15(1):1-25. doi:10.1002/hbm.1058
11. Friedman L, Glover GH, Krenz D, Magnotta V, FIRST BIRN. Reducing inter-scanner variability of activation in a multicenter fMRI study: role of smoothness equalization. *Neuroimage*. 2006;32(4):1656-1668. doi:10.1016/j.neuroimage.2006.03.062
12. Bowring A, Maumet C, Nichols TE. Exploring the impact of analysis software on task fMRI results. *Hum Brain Mapp*. 2019;40(11):3362-3384. doi:10.1002/hbm.24603
13. Brett M, Johnsrude IS, Owen AM. The problem of functional localization in the human brain. *Nat Rev Neurosci*. 2002;3(3):243-249. doi:10.1038/nrn756

REVIEWERS' COMMENTS

Reviewer #2 (Remarks to the Author):

The authors have made considerable efforts to improve the manuscript and have been responsive to the reviewers' concerns wherever possible. Many aspects of the manuscript have been clarified in terms of both the text and the figures, as well as additional supplementary analyses. The major claims, conclusions, and limitations are much more clear so that it can be effectively concluded that the novelty and information provided from this paper will be beneficial to the field of neural mechanisms of placebo and pain.